# Machine learning methods to assess the effects of a non-linear damage spectrum taking into account soil moisture on winter wheat yields in Germany

Michael Peichl [1], Stephan Thober [1], Luis Samaniego [1], Bernd Hansjürgens [2], and Andreas Marx [1]

[1]UFZ-Helmholtz Centre for Environmental Research, Department Computational Hydrosystems, Permoserstrasse 15, D-04318 Leipzig, Germany
[2]UFZ-Helmholtz Centre for Environmental Research, Department Economics, Permoserstrasse 15, D-04318 Leipzig, Germany

**Correspondence:** Michael Peichl (michael.peichl@ufz.de), Andreas Marx (andreas.marx@ufz.de)

**Abstract.** Agricultural production is highly dependent on the weather. The mechanisms of action are complex and interwoven, making it difficult to identify relevant management and adaptation options. The present study uses random forests to investigate such highly non-linear systems for predicting yield anomalies in winter wheat at district level in Germany. In order to take into account sub-seasonality, monthly features are used that explicitly take soil moisture into account in addition to extreme

meteorological events. Clustering is used to show spatially different damage potentials, such as a higher susceptibility to drought damage from April to July in eastern Germany compared to the rest of the country. The variable that explains most differences is soil moisture in March, where higher soil moisture has a detrimental effect on crop yields. In general, soil moisture explains more yield variations than the meteorological variables, while the top 25 cm of soil moisture is a better yield predictor than the total soil column. The approach has proven to be suitable to explain historical extreme yield anomalies

for years with exceptionally high losses (2003, 2018) and gains (2014) and the spatial distribution of these anomalies. The highest test R-square is about 0.70. Furthermore, the sensitivity of yield variations to soil moisture and extreme meteorological conditions, as shown by the visualisation of average marginal effects, contributes to the promotion of targeted decision support systems.

## 1 Introduction

Extreme weather conditions have increased over the last two decades over Germany, leading to an amplification of inter-annual crop variations in the agricultural sector. These include years with above-average wet years (2002, 2007, 2010), but also the droughts of 2003, 2015 and 2018 and the year 2012 with a longer period of bare frost (Gömann, 2018). Models that accurately map weather conditions to crop yields allow a better understanding of the damage mechanism and can thus support management and adaptation (Albers et al., 2017; Peichl et al., 2018) as well as be used for decision support systems

and seasonal forecasts (van der Velde et al., 2019; Lecerf et al., 2019; Sutanto et al., 2019; Ben-Ari et al., 2018; Guimarães Nobre et al., 2019). Furthermore, such damage functions form the basis for projections of the social and economic effects of climate change (Carleton and Hsiang, 2016; Diaz and Moore, 2017; Hsiang et al., 2017). While process-based crop models take





into account the growth mechanisms of crops (Rosenzweig et al., 2014), they are only partially able to reproduce historical yield anomalies (Müller et al., 2017; Mistry et al., 2017). Furthermore, it has been shown that classical statistical models

outperform process-based models in predictive power, especially on a large scale (Lobell and Asseng, 2017). Those statistical approaches usually reduce the processes that affect plant development to the main features (Timmins and Schlenker, 2009; Kolstad and Moore, 2020). Following the seminal work of Schlenker and Roberts (2009), extreme heat is routinely included as the main determinant (Carleton and Hsiang, 2016). However, we consider inference about the marginal effect of these often aggregated measurements of meteorological variables on the yield to be critical, since a spurious association can be

caused by missing or only roughly represented variables (Peichl et al., 2018; Roberts et al., 2017). For example, a global study based on process-based models for maize and wheat found that for most countries water stress is a major source of the observed yield variations (Frieler et al., 2017). It has also been shown that it is necessary to account for multiple adverse environmental conditions such as frost, heat, drought and excessive soil moisture during sensitive growth phases (Trnka et al., 2014; Albers et al., 2017; Schauberger et al., 2017; Mäkinen et al., 2018; Peichl et al., 2018, 2019). Furthermore, these effects

are often mutually amplifying, which potentially increases the impact (Ben-Ari et al., 2018; Lu et al., 2018; Toreti et al., 2019; Zscheischler et al., 2018, 2020). In 2018, for example, extremely hot temperatures in Germany were accompanied by extremely low precipitation, which further intensified the effects on crop yield (Zscheischler and Fischer, 2020). Ben-Ari et al. (2018) showed that compound extreme events such as exceptionally warm temperatures in late autumn and very wet conditions in late spring 2016 led to unprecedented wheat losses in France.

In previous studies we have tried to approximate this non-linear and complex damage spectrum by considering the sub-seasonal effects of hydro-meteorological variables such as temperature and soil moisture, however, applying an econometric linear model neglecting sub-seasonal interaction of the features. This approach was very well able to project long-term mean yield changes, but not the inter-annual variations caused by extreme conditions (Peichl et al., 2019). This study applies a statistical framework that takes into account a range of potentially harmful extreme environmental conditions. For this purpose

we map various sub-seasonal hydro-meteorological extremes with yield anomalies of winter wheat. For winter wheat, the challenge of nonlinearty is particular relevant: studies have shown that it is difficult to explain yield variations in winter wheat because the growing season is relatively long compared to other crops (Vogel et al., 2019). In accordance with the typology of compound weather and climatic events (Zscheischler et al., 2020), we consider plant growth as a nonlinear system, since the time of occurrence and the various features and extreme events themselves interact, which ultimately affect plant development

(Storm et al., 2020). Therefore, we use random forests, which is a machine learning algorithm particularly suitable for complex nonlinear systems with interactions in the predictors (Breiman et al., 1984; James et al., 2013; Vogel et al., 2019). The features used (see Table 1) are meteorological extreme indicators for temperature and precipitation extremes as well as soil moisture, which is the main water source for plant growth, each on a monthly basis. This allows for sub-seasonality in the model and the quasi-consideration of plant growth and different phenological stages. To increase the predictive power (Conradt et al., 2016)

of the models as well as to reveal spatially dependent damage mechanisms, we rely on spatial clustering, which accounts for regional differences in climate, soil moisture and soil properties.





Disentangling the non-linear spectrum of extreme conditions harmful to plant growth and identifying the causes of yield loss will help improve decision support systems in the agricultural sector. Machine learning focuses primarily on predictive accuracy, while econometricians focus on inference, i.e. deriving statistical properties of estimators for hypothesis testing

within a classical parametric and linear approach (Mullainathan and Spiess, 2017; Storm et al., 2020). However, the functional forms used in econometric analysis are usually not flexible enough to capture the interactions, non-linearities and heterogeneity that are often common to both biological and social processes in agricultural and environmental systems (Storm et al., 2020). On the other hand, there is concern from an econometric point of view that machine learning models are difficult to interpret because of these high-dimensional and highly non-linear functions (Breiman, 2001b; Zhao and Hastie, 2019). To address

this issue of interpretability, we also present the significance of the variables and the mean average effects represented by Accumulated Local Effect Plots (Apley and Zhu, 2016) of the main characteristics for each cluster. The paper describes the data (Section 2), methods (Section 3) and results (Section 4). Most results are discussed in the results section. A short conclusion is given at the end.

## 2   Data

The annual yield data for winter wheat are provided by the Federal Statistical Office for the counties from 1999 to 2018 (Statistische Ämter des Bundes und der Länder, 2019). Winter wheat has the largest share in cultivated area (2018: 46 %) and total production (2018: 51 % of quantity harvested) (Statistisches Bundesamt (Destatis), 2018) amongst all crops in Germany. Figure 1a shows a map of the average yield and the standard deviation for the period 1999 - 2018. On average, the highest yields are recorded in the extreme north of Germany, while the lowest yields and the highest inter-annual variation are found in

the eastern part of Germany. For each county, the data is converted into yield anomalies in percent by subtracting the average yield and dividing the resulting difference by this average. We have not corrected the yield data for the trend in order to take for example technological developments into account. Since the mid-1990s, annual yield increases have stopped and no trend in yields has been observed since then (Gömann, 2018). This is shown in Figure 1b, which shows the distribution of yield anomalies for the period 1999 - 2018. A positive linear trend can be observed for this time period (blue line). However, as

can be seen from the green line, which represents the fit of the Local Polynomial Regression, this positive linear trend is mainly associated with the above-average yields from 2013 onwards, which first rise rapidly and then fall again. Accordingly, almost no linear trend can be observed for the years before that (orange line in Figure 1b). A trend correction is therefore not necessary. All counties with yield data of less than ten years of observations are removed from the analysis, which results in 350 remaining districts (figure A1 in the appendix shows a map of the numbers of observations available for each county).

The daily temperature and precipitation data are obtained from a network of stations of the German Weather Service (Deutscher Wetterdienst, 2019). For the interpolation method to gridded data see Zink et al. (2017). Daily meteorological data are converted into monthly aggregates by counting the days above or below a defined threshold based on Gömann et al. (2015). Table 1 shows the seven meteorological extreme indicators, the underlying meteorological variables and considered months as well as the corresponding variable names in the model.



**(a)**

**(b)**

**Figure 1.** (a) 20-year winter wheat yield average (1999-2018, left) and standard deviation (right) of yields for the counties over Germany. (b) Box-and-whisker plots of winter wheat anomalies for each year and both linear and nonlinear model fits to identify potential trends in the anomaly data. Exceptional years of interest are marked with a light beige box. Data source: Federal Statistical Office DESTATIS



**Table 1.** Table of the indicators of seven extreme weather conditions as well as the according meteorological conditions and months of occurrence. The indicators (first column) are generated by counting the days above or below the thresholds of certain meteorological variables for specific months (second column). The variable names of the resulting features are displayed in the last column. The number indicates the month. For example, Frost10 represents the number of days with black frost in October of the previous year, and Heat6 the number of days with heat in June. T reflects temperature, P precipitation.

| Ext. weather conditions | Meteorological variables | Variable Names |
|---|---|---|
| Black Frost | min. T $< -20°$C: Dec. - Feb. | Frost12, Frost1, Frost2 |
| | min. T $< -10°$C: Mar. & Nov. | Frost3, Frost11 |
| | min. T $< -5°$C: Oct. | Frost10 |
| Late Frost | min. T $< 0°$C: May | Frost5 |
| Alternating Frost | min. T $< -3°$C min. T $> 3°$C: Jan. - May | AF1, AF2, AF3, AF4, AF5 |
| Heat | max. T $> 30°$C: Apr. - Aug. | Heat4, Heat5, Heat6, Heat7, Heat8 |
| Heavy rain season | P $> 30$ mm/d: Oct. - Jun. | Rain10, Rain11, Rain12, Rain1, Rain2, Rain3, Rain4, Rain5, Rain6 |
| Rain harvest | P $> 5$ mm/d: Jul. & Aug. | Rain7, Rain8 |
| Precipitation scarcity | P $= 0$ mm/d: Oct. - Aug. | PS10, PS11, PS12, PS1, PS2, PS3, PS4, PS5, PS6, PS7, PS8 |

90 The soil moisture simulation was obtained from the German Drought Monitor (Zink et al., 2016) using the mesoscale Hydrologic Model (mHM) (Samaniego et al., 2010; Kumar et al., 2013). In general, the model is grid-based with a spatial resolution of 4 km. Various hydrological processes such as infiltration, percolation, evapotranspiration, snow accumulation, groundwater recharge and storage, and runoff, both rapid and slow, are considered to calculate soil moisture. The model is driven by hourly or daily meteorological forcings (e.g., precipitation, temperature). For parameterization, it uses the spatial variability

95 of observable but high-resolution physical properties of the catchment (land surface descriptors such as the digital elevation model, slope, aspect, rooting depth based on land cover classes or plant functional types, plant canopy characterization, soil texture, and geological formation type). The main feature here is the multiscale parameter regionalization, which is critical to achieve cross-scale flow matching. It allows the derivation of seamless parameter arrays between the targeted resolution and the high-resolution land surface descriptors (Samaniego et al., 2017). However, no endogenous land use management processes

100 are considered. The depth of the soil in each grid depends on the soil type used in mHM.

 Soil moisture is presented here as an index because an index configuration supports the reduction of systematic errors associated with, for example, simulation or spatial processing (Auffhammer et al., 2013; Lobell, 2013). The soil moisture index (SMI, Samaniego et al. (2013)) is derived from a non-parametric and site-specific cumulative distribution function of soil moisture for the period 1951-2019 for each month of the vegetation period of winter wheat. The percentile-based index

105 thus quantifies the likelihood of occurrence of the monthly absolute soil moisture. The index ranges from zero to one and





represents an anomaly with respect to the monthly long-term soil water median (SMI = 0.5). Low values represent extremely dry soils and high values represent extremely wet soils. Consequently, seasonal effects due to drought and wet conditions during different agrophenological stages are taken into account. In this context, the interpretation of the monthly indices must take into account that the proportion of saturated soil changes over time and thus the base value for the index of each month.

For Germany, this seasonality of soil moisture is shown in Fig. 4 in Samaniego et al. (2013). Here, we include two variables denoting soil moisture at two depths, namely the uppermost 25cm (SMI) and the total soil column (SMIa) with variable depth depending on the soil map BUEK1000 (BGR, 2013). Soil moisture is highly persistent in time, providing an integrated signal of meteorological conditions in the preceding and following months (, e.g.) Orth2012, Samaniego2013a. This autocorrelation does not allow for cumulative measures as with temperature or other meteorological extremes but allows the use of monthly

averages. Because of the high positive time correlation of soil moisture that accounts for the entire soil to its first- and second-order neighbors, only October, January, April, and July are considered for SMIa (Figure A2, Appendix). The yield data are available for the counties of Germany. The meteorological data and soil moisture, which have a spatial resolution of $4 \cdot 4 \ \mathrm{km}^2$, are thus aggregated to the counties. See Peichl et al. (2018) for a detailed description of the spatial processing (e.g., grid cells are masked for non-irrigated agricultural land).

## 3 Method

We apply the machine learning method Random Forests to explain the variation of winter wheat anomalies by the hydro-meteorological features introduced above. Random Forests (RFs) have been used to analyze the effect of meteorological determinants on crop yields on a global scale (Jeong et al., 2016; Vogel et al., 2019) and in specific countries or regions (Jeong et al., 2016; Hoffman et al., 2018; Beillouin et al., 2020). However, none of these approaches explicitly used a measure of soil

moisture nor did they apply clustering to take into account the region-specific yield potential. RFs have also been widely used in related disciplines such as drought impact assessment (Bachmair et al., 2016) and forecasting (Sutanto et al., 2019). Within these applications, it has proven to be more powerful for classification than other data-science methods (Bachmair et al., 2017). Here, for a domain covering the whole of Germany, RFs proved to be superior to other machine learning algorithms that are particularly suitable for nonlinear systems, such as support vector machines and gradient boosting (not shown). This result is

in alignment with other studies on global scale (Vogel et al., 2019). An comparison of RFs to Least Absolute Shrinkage and Selection Operator in the Northern Hemisphere showed comparable for a binary classification approach for simulated crop failures (Vogel et al., 2020). A RF randomly produces numerous independent trees as an ensemble to avoid over-fitting and sensitivity in the configuration of training data, while being very efficient (Sutanto et al., 2019). The trained model is Breiman's RF (Breiman, 2001a). It is tuned to the number of variables available for splitting at each tree node (parameter mtry) using

the tuneRF function of the R package randomForest (Liaw and Wiener, 2002). The initial values of the parameters are set to default, the number of trained trees is 500, and the tuning is based on an out-of-bag error estimation (see e.g. James et al. (2013) for more information).





The crop yield potential varies regionally in Germany due to differences in climate and soils among other factors. To take account of these differences, a spatial clustering was implemented to identify different subregions within Germany. The clus-
tering methods used are representatives of centroid-based ones, such as k-means (KMEANS, (MacQueen, 1967; Hartigan and Wong, 1979)) and partitioning around medoids (PAM, (Kaufman and Rousseeuw, 1990)), which is less sensitive to outliers, as well as the connectivity-based hierarchical clustering (HIERARCHICAL, (Murtagh, 1985)). Standard internal validation such as connectivity, average silhouette width, Dunn index for cluster numbers between 2 and 16 were tested for the evaluation. However, the results show no clear outcome on which algorithm and size combination to use (figure A3). Instead, we fit the
random forests individually for each region defined by one of the cluster algorithms and cluster size. We then selected the combination that maximizes the average prediction capacity (test R-square) across all regions. For each of these cluster con- figurations the model is trained on 80 percent of the data in that subset and predicted for the rest. The data used for clustering are monthly averages and daily observations of the meteorological data for the entire year. SMI is included for both the upper layer and the entire soil column. Average yields are also taken into account in the data for cluster formation. This is based on
the intuition of taking into account time-invariant factors of each cluster that affect yields such as soil quality and average farm size. These factors are not considered in the random forest due to use of yield anomalies. This approach is inspired by fixed effect econometric models. There, the group means are fixed, thus taking into account the time-invariant heterogeneity of these groups (for econometric literature see for instance Wooldridge (2012).

The random forest algorithm allows to study the relationships between hydro-metrological extremes and yield anomalies
by assessing the relative importance of the variables and the functional relationship between each predictor and the response variable (Jeong et al., 2016; Vogel et al., 2019; Beillouin et al., 2020). Furthermore, we use model agnostics, which includes various flexible methods that allow the interpretation of black box models that separate the explanation of the model from the model itself. Accordingly, the same method can be used for any kind of machine learning algorithm, different types of explanations and different types of features can be presented (Ribeiro et al., 2016). The particular method considered here is
Accumulated Local Effects (ALE), which is a visualization of the average marginal effect of features on target variables for supervised learning models (Apley and Zhu, 2016; Molnar, 2020). ALE plots predict the effect of an explanatory variable across their realisations, taking into account only a subset of the sample with observed values adjacent to the respective realisation (Apley and Zhu, 2016). It is a faster alternative to the popular approach of the Partial Dependence Plots (Friedman, 2001), which have already proved to be suitable in the context of yield prediction (Jeong et al., 2016; Vogel et al., 2019). However,
ALE plots are more suitable to visualize marginal effects by plotting explanatory variables against the predicted outcomes if the features are highly correlated (Storm et al., 2020). One limitation is that uncertainty estimates are not provided for ALE plots, which is a substantial limitation of the approach and is an area of active research (Storm et al., 2020). The ALE plots are shown for the most important features of each cluster.





**Table 2.** Table with the average R-square (test) for the three best combinations of cluster algorithm and cluster size (in parentheses) for three soil moisture configurations.

| Soil moisture configuration | Algorithm size combination | Avg. R-square (test) |
|---|---|---|
| SMI for uppermost 25cm | KMEANS (8), PAM (8) | 0.70 |
| | PAM (3) | 0.69 |
| | PAM (2) | 0.69 |
| | non-cluster | 0.65 |
| SMI for entire soil column | KMEANS (10), PAM (8) | 0.68 |
| | PAM (8) | 0.67 |
| | HIERARCHICAL (6), KMEANS (6) | 0.67 |
| | non-cluster | 0.64 |
| SMI for both uppermost 25cm and entire soil column | KMEANS (8), PAM (8) | 0.69 |
| | PAM (2) | 0.69 |
| | KMEANS (10) | 0.68 |
| | non-cluster | 0.65 |

## 4 Results

### 4.1 Evaluation of spatial clustering

To evaluate the cluster algorithm and the number of clusters the test R-squared for each cluster and number of cluster combination is generated. Table 2 shows results for three different soil moisture configurations, i.e. one each for the upper layer as well as the entire soil column and one that takes both into account. For each of these soil moisture configurations the three combinations of algorithms and cluster sizes with the highest R-square are shown. The validation criterion for non-cluster formation is also shown as a reference. Overall, the best results can be achieved if only SMI for the uppermost 25 cm is considered. The best results explain 70% of the wheat yield anomaly variation. The average variance explained exceeds the variance explained by RFs applied at the global level (Vogel et al., 2019) or for the Northern Hemisphere (Vogel et al., 2020) and is comparable to the highest explained variability for RFs applied to European regions , with an average for winter wheat of 43% (Beillouin et al., 2020). A comparable regression model approach is able to explain a maximum of 32% of the variation (Gömann et al., 2015). A large fraction of the variability is usually explained by time-invariant factors, which are largely not considered here due to the demeaned yield data. For example, Peichl et al. (2018) using a regression model for silage maize showed that up to 32% of the variation explained by the model is explained by time-invariant factors. An approach modelling relative year-to-year yield changes has similar results (Conradt et al., 2016). There, the best explanatory power is found for northern and eastern Germany with comparable coefficients of determination. However, for the rest of Germany the model presented here performs better as it is doing well in regions with rather low yield variability such as in the south of cluster 4 or in Bavaria (figure 2a).





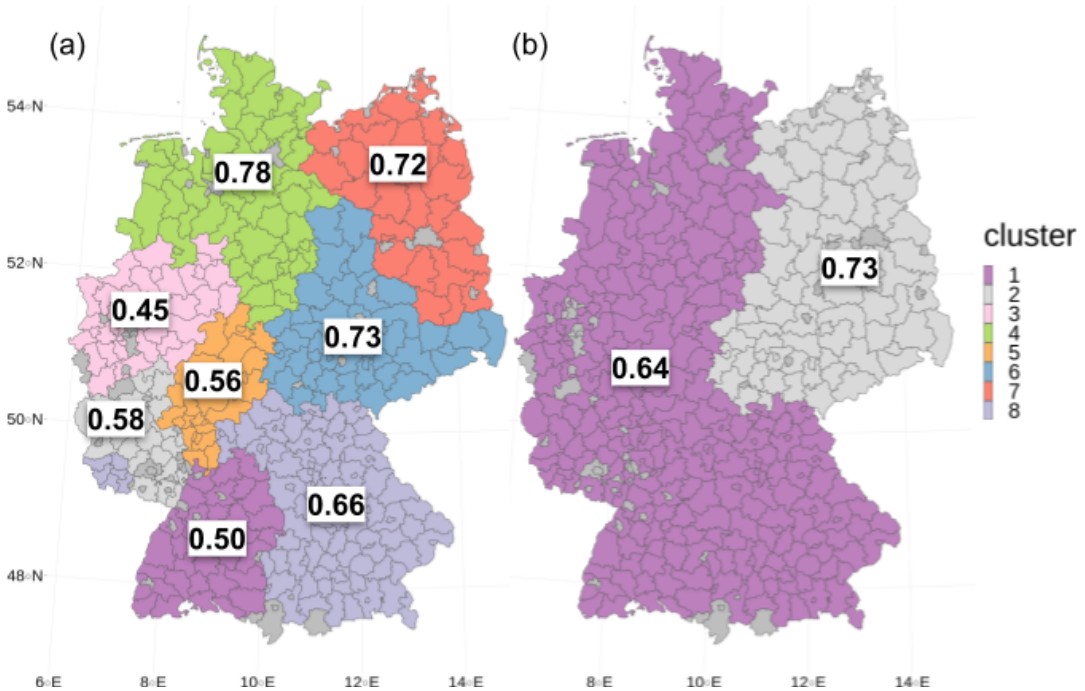

**Figure 2.** Spatial structure of clusters derived from the PAM algorithm with 8 clusters (a) and with 2 clusters (b).

We choose to further explore the results of PAM with two clusters because it provides a compromise between a high predictive power and reduced complexity. The clusters are divided along the former border between western and eastern Germany (see figure 2b). This division along administrative borders is also supported by other cluster (figure 2a). In general, a higher variability of yields but lower average yields can be observed in most parts of eastern Germany (figure 1). This indicates structural differences between western and eastern Germany (Albers et al., 2017). In addition to environmental conditions, these may include, for example, different data collection practices and different agribusiness structures. The R-square of cluster 1 is 0.64 and of cluster 2 is 0.73. For both clusters, a good fit can be observed in the scatter plots for most of the data (figure 3 (a)), while the tails are slightly underestimated (figure 3 (b)). The higher explanatory power for cluster 2 might be related to the higher variation in yield anomaly there (see figure 1). Furthermore, different impact mechanisms might work within each cluster. Those are disentangled in the following.

## 4.2 Marginal effects of the most important features

The main variables for each sub-cluster and the corresponding average marginal effects are presented below in order to understand the range of adverse effects on yield variation in winter wheat. To generate variable importance and ALE plots, no split is made between test and training data. The non-cluster results are compared with the spatial clusters generated with the PAM clustering algorithm for a cluster size of 2 (PAM (2)). The detailed ALE plots for the overall best algorithm cluster size



**Figure 3.** Scatter (a) and density (b) plots of the observed and predicted data for the two clusters derived using the PAM algorithm with size 2.



combination (PAM (8)) can be found in the appendix (figure A4). In general, the ALE visualization there is more wiggly, which indicates an over-fit of the model. The effects shown here are additive as the they are cleared off the correlation to other features.

The ALE plots in figure 4 are ranked in accordance to their variable importance (for further information see the variable importance section in the appendix). In general, soil moisture supports best the performance of the model. This is valid in particular for the non-cluster approach and cluster 1, since in cluster 2 more meteorological variables are critical. Figure 4a shows the ALE plots for the non-cluster approach. Cluster 1 (figure 4b) shows almost similar sensitivities to those observed for the non-cluster approach. Besides the importance ranking, basically only the amplitude of the effects change. Only the least important variable is PS5 instead of PS4 in cluster 1. Topsoil SMI in February and March as well as in August show a positive signal for shortage in soil moisture. This indicates a preference for drier than normal conditions during those months. For December and January, no negative impact of soil water scarcity can be found. SMI shows a negative drought signal for the months April to July with the one in June being the largest. For July a drop in yield can be observed for small values of SMI. However, the negative effects of soil water abundance are much larger in this case.

For cluster 1, in May and June, the drought signal by soil moisture is comparatively smaller than the one found in the non-cluster setting. For April the signal is stronger but still ambiguous. Overall, the signals associated with water deficit stress are rather weak in both cluster approaches, but particularly in cluster 1. This is consistent with the results of a statistical model for North Rhine-Westphalia, which compromises a large part in the west of cluster 1, according to which water stress has no limiting effect on wheat yield there, not even due to climate change (Kropp et al., 2009). In cluster 1, for most of the ten soil moisture variables more than normal water is comparatively more harmful than water shortage in the soil. This is partially consistent with the results of a previous study that showed higher sensitivity of wheat yields to excess water compared to drought, but for Germany as a whole (Zampieri et al., 2017).

However, this result of Zampieri et al. (2017) does not hold for the sub-region defined by cluster 2, where the observed pattern in feature effects changes (Figure 4c). There, for the months May to July a lower than normal soil water level is much more harmful than to much water in the soil. This strong negative water deficiency signal is visualized for the period typically associated with the drought-sensitive vegetative and generative phases of winter wheat (Lüttger and Feike, 2018). For example, expected wheat yields in June and July fall by about 5 and 4 percent, respectively, for each month with an SMI value lower than 0.125. For the SMI features, a pivotal transition in the patterns takes place between April and May, as the negative effects of drought are evident first in May. Interestingly, water abundance in January (SMI larger than 0.75) is associated with much higher losses compared to cluster 1 and the non-cluster setting. Counter-intuitive effects can be observed for April: fewer days without rain are advantageous (PS4), but a positive effect of soil moisture drought can be observed (SMI4). This could be a result of the persistence of soil moisture compared to precipitation-only measurements due to the water holding capacity of the soil. In contrast to the other clusters, June shows a minor positive effect of higher than normal soil water on crop yield, but no negative effect for high SMI values that account for excessive water supply. In July and August, the effects of soil water levels much higher than normal are similar to the other clusters.

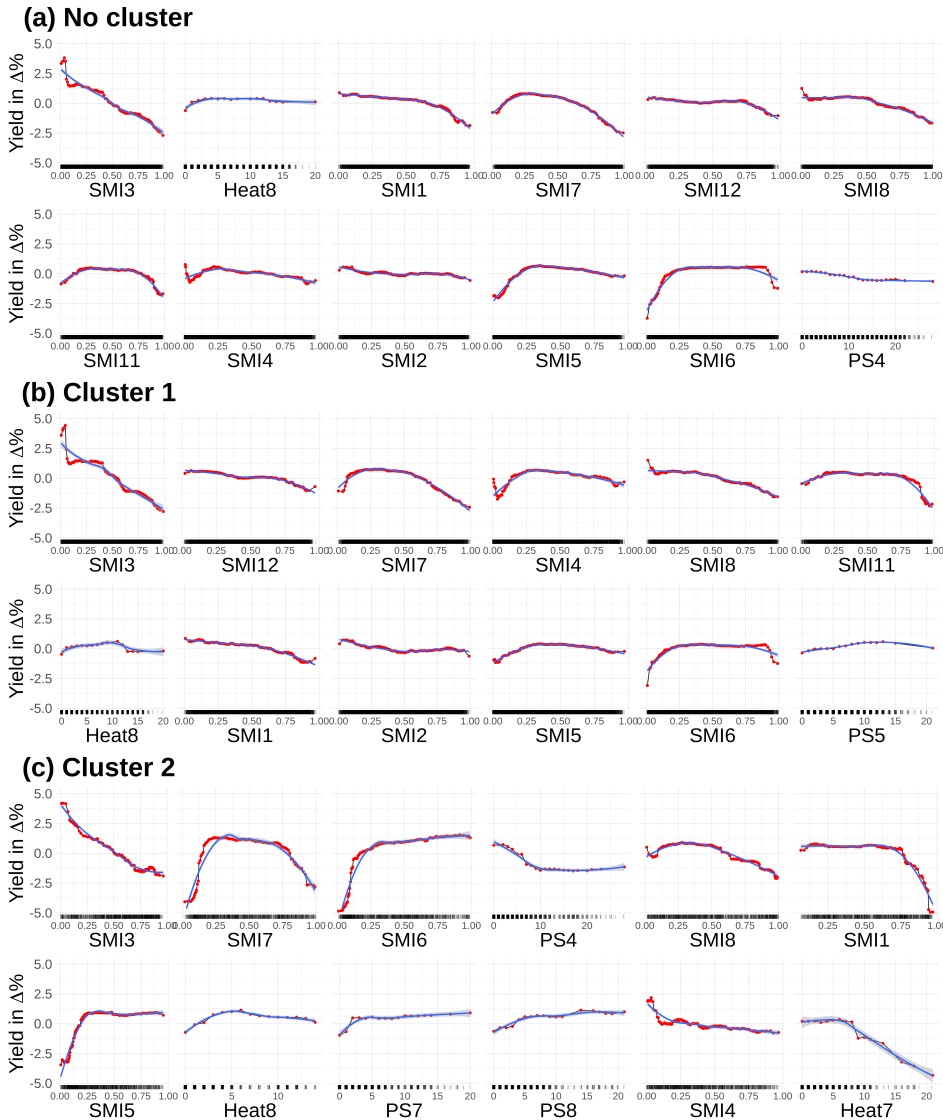

**Figure 4.** Accumulated local effects plots of the twelve most important features for no cluster (a), cluster 1 (b), and cluster 2 (c). The red dots are estimated by the ALE plot algorithm (FeatureEffects of the iml-package in R). We have chosen a rather large interval size of 100, which allows us to reveal the true complexity of the model at the expense of shakiness. Therefore a nonlinear smoothing function (LOESS - locally estimated scatterplot smoothing) is added in blue (with confidence interval in grey). SMI represents the soil moisture index for the uppermost 25 cm of the soil column, PS stands for days without rain in a given month and Heat for days with a maximum temperature of more than 30 degrees. The number indicates the month, 10, 11, and 12 refers to the year before. For example, SMI10 represents SMI in October, i.e. the start of the growing season.





In our study, the most important variable for all three cluster considerations is soil moisture in March. The relationship between the SMI in March and yield anomalies is generally negative for the entire SMI. This indicates that yield losses are associated with higher than normal water content in the upper 25 cm of soil. This finding is consistent with the results of Ben-Ari et al. (2018), which showed that a combination of abnormally wet conditions together with abnormally warm temperatures (not controlled in this study for) in late fall leads to large losses in winter wheat in France in 2016. An evaluation of the
interaction effects to treat possible compounding events does not show stable results and varies from run to run, probably due to the lack of available data. Therefore these results are not discussed here but needs to be evaluated in further studies.

A strong drought signal can only be found in the data if the model is applied to a sub-region of Germany, here eastern Germany. This sub-region is defined by a clustering approach that particularly accounts for different crop potentials. In a non-cluster approach, those signals are mostly confused. This underlines the importance of using clustering to take account
of different crop potentials and environmental conditions. The observation that the absence of water govern crop production in this eastern Germany is in alignment with recent studies (Conradt et al., 2016; Vinet and Zhedanov, 2010). There, lack of precipitation together with sandy soils, which have a lower water holding capacity, may result in water shortage for winter wheat growth (Rezaei et al., 2018). For the rest of Germany for most growing stages, extensive wet periods with water-saturated soil represent an extreme weather situation for agriculture (Gömann, 2018). The most sensitive growth phase for waterlogging
is after germination, but before emergence (Barber et al., 2017; Grotjahn, 2020). Oxygen deficiency can cause damage to the plant that result in yield losses (Cannell et al., 1980). In addition, excessive soil water fosters pathogens (Grotjahn, 2020) and complicates plant treatment operations (Urban et al., 2015; Gömann, 2018).

Generally, it is difficult to disentangle the compounding effects of heat and water supply on plant growth (Gourdji et al., 2013; Roberts et al., 2013; Lobell and Asseng, 2017; Roberts et al., 2017; Schauberger et al., 2017; Siebert et al., 2017; Zscheischler
and Seneviratne, 2017; Mäkinen et al., 2018; Peichl et al., 2018). Previous research shows that the specific contributions of temperature and precipitation anomalies to drought are difficult to isolate (Zscheischler and Seneviratne, 2017; Vogel et al., 2019). Furthermore, the negative yield effects of high temperatures are associated with water stress and can be mitigated by irrigation (Frieler et al., 2017; Vogel et al., 2019; Ribeiro et al., 2020). However, for Germany studies show that heat was more harmful than drought during sensitive growing stages in Germany in the past (Lüttger and Feike, 2018; Trnka et al., 2014).
Vogel et al. (2019) showed on a global scale using a very similar approach that temperature-related indicators such as the frequency of warm days, the average temperature of the growing season and the average diurnal temperature have the highest predictive power for crop yields. Here, however, neither for cluster 1 nor cluster 2, a heat signal is observed for June, which is associated with the most heat sensitive phase of anthesis (Barber et al., 2017; Rezaei et al., 2018). In cluster 2, more than 8 days of heat above 30 degrees in July show adverse effects, a period that could be linked to grain filling (Lobell et al., 2012;
Lüttger and Feike, 2018; Mäkinen et al., 2018). In both clusters, heat in August, a period generally associated with ripening, has positive effects for each additional day and from day 11 onward negative effects. Our approach, which explicitly controls for the water supply of plants by soil moisture, shows more water-related effects compared to heat effects. This indicates that especially for East Germany the water deficit in the upper 25 cm of the soil is a more relevant factor than heat. This has implications for management and adaptation measures, such as an adequate irrigation infrastructure instead of heat-resistant





### 4.3    Predictions of years with extreme yield anomalies

Figure 5 shows the maps of observed, the predicted and the difference between those two for winter wheat yield anomalies for the years in-sample years 2003, 2014 and 2018 as well as for 2019. The first three are the years with both the largest losses and gains during the training period. Those years show different spatial pattern in yield gains and losses. In 2003, the year with the highest total volume of losses, the largest losses were recorded in eastern Germany. For the year 2018 the losses are more likely to be in the northernmost parts of Germany, while the south of Germany shows positive yield anomalies. 2014 is a particularly good year with higher than expected yields, especially in the easternmost parts of Germany. The general spatial patterns of losses and gains of the observed data are represented by the simulated data for all three years. However, as can be seen from the differences, the model tends to slightly underestimate the extent of both extremes. For example, the largest negative differences between observed and projected data for 2003 are found for Vorpommern-Greifswald, a county in the north-east of Germany. The region around this county also shows the largest contiguous area of negative differences, i.e. an underestimation of the losses. The largest positive difference is found in the very south. For 2018 the picture is comparable and the positive yield anomalies in the south and the negative anomalies in the north are underestimated. For both years, however, there is no clear pattern of over- and underestimation for estimating values between the two extremes. For 2014, the very positive results in the easternmost districts are underestimated. However, the highest positive differences are not consistent with the highest positive anomalies observed. The highest differences in the positive anomalies are those for the high yield anomalies in the extreme southwest. The negative differences are for the underestimated losses in southern Bavaria. For the in-sample years 2003, 2014, and 2018 the model is very well able to predict district yield anomalies, but does not represent the full extent of the anomaly variation in the extremes. With less variation in the observed yield data, no clear pattern of under- or overestimation can be observed. A different picture can be observed for the out-of-sample year 2019. There, both losses and gains are structurally underestimated and the full range of variation of the observed yield anomalies is not represented in the predicted yield anomalies. This shows potential difficulties of out-of-sample predictions of machine learning models such as random forest. One possible reason for this is that out-of-sample structural relationships and functional relationships of a given year are not detected in this approach because they do not occur in the sample. Corresponding patterns affecting wheat yields in 2019 might not have occurred in the 1998 - 2018 training period. For example, damage events due to certain determinants, compounded or isolated, could have occurred in 2019 that are not represented in the model.

## 5    Conclusions

Here we show that random forests are very suitable for assessing the non-linear damaging effects of different environmental conditions on winter wheat yield anomalies. Explicit consideration is given to soil moisture at various depths. In addition, the crop potential and other spatially related environmental conditions are taken into account, which helps to improve predictive



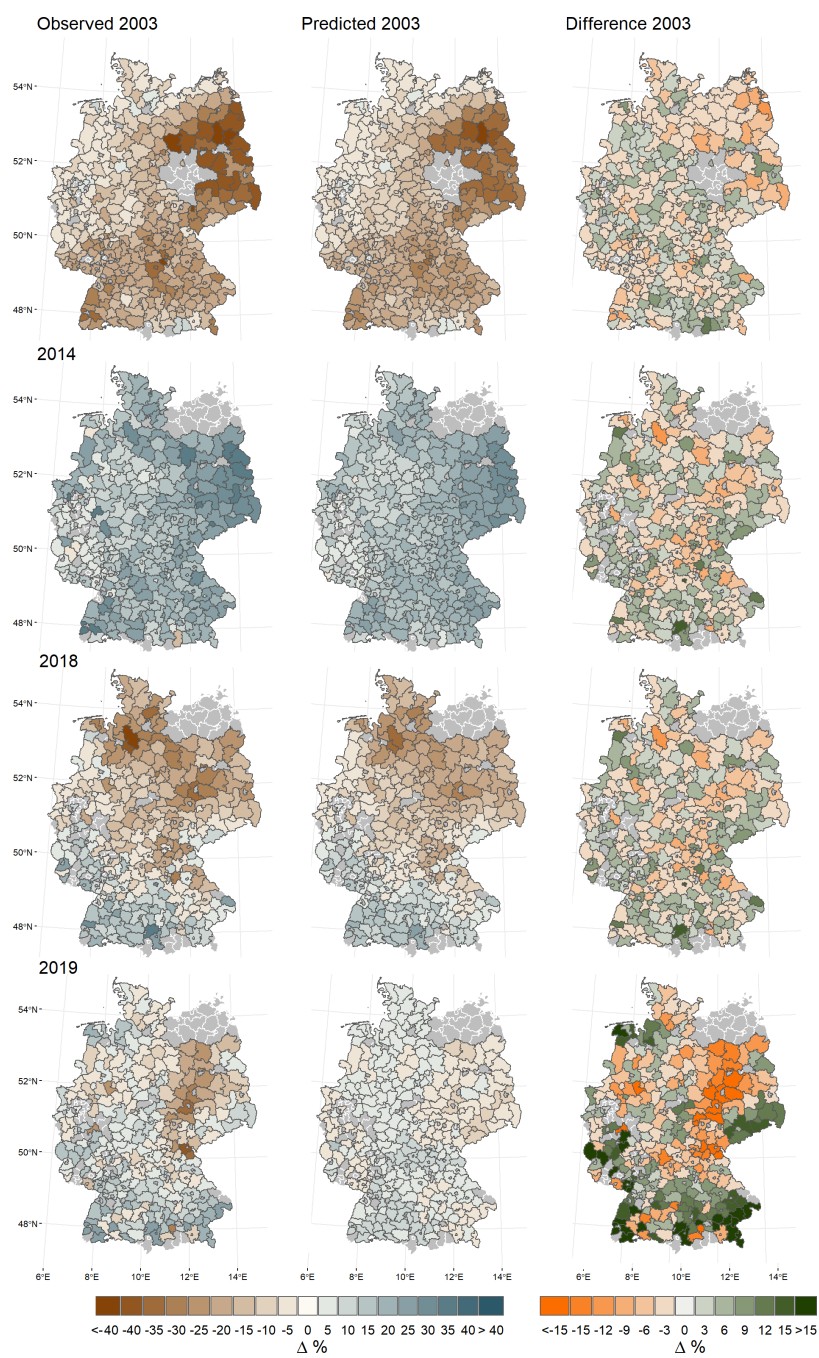

**Figure 5.** Maps of the observed, the predicted (for KMEANS/PAM clustering with 8 subregions) and the difference between these two for winter wheat yield anomalies for the in-sample years 2003, 2014 and 2018 as well as 2019, for which the model was not trained, at the county- level.





power. Different clustering algorithms and cluster sizes have been applied to improve the predictive capacity of the model from 65% in average test R-squared to 70%. In general, the approach is able to explain the general pattern of losses and gains of the counties, also those in particular extreme years such as the years 2003, 2014, and 2018. In comparison to other models, this

approach performs better in regions with low crop yield variance. However, it slightly underestimates the extremes. This issue is more prominent for out-of-sample predictions. This suggests that the out-of-sample predictive capacity of machine learning algorithms such as Random Forest needs to be further explored both for use as a seasonal forecasting tool and in the context of climate impact assessment. Nevertheless, the analysis presented here can support the design of tailor-made and, above all, prompt support mechanisms for large losses caused by extremes as it helps to disentangle the damage spectrum for sub-regions

in Germany. It particularly shows, that soil moisture dominates the variable importance ranking. Thereby, it is shown that it is preferred to account for the upper 25 cm of the soil moisture column compared to the entire column or a combination of both. All over Germany, soil moisture abundance in March thereby ranks first and shows the most substantial negative effects. In addition, the abundance of water is problematic for the growth of winter wheat in most other parts of Germany. Only the northeastern part of Germany is rather driven by damages related to water shortages. Those water shortage effects

for the smaller cluster remain mostly undetected in a non-cluster approach. Meteorological variables, such as heat-related measures, are comparatively low ranked in explaining the impact on yield anomalies in winter wheat. Only in eastern Germany a relevant negative heat effect can be observed, which, however, ranks 12th among all variables considered. Those information are helpful to tailor management and adaptation measures. For example, it is particularly suitable for the insurance industry to provide index-based insurance policies, as they help to identify harmful features and visualize thresholds in those features that

cause damage (Albers et al., 2017). Prominent examples of this are the large yield declines associated with an SMI smaller than 0.125 in June and July in eastern Germany. These sub-seasonal thresholds may also help to better determine drought classes for specific crops used in monitoring and decision support tools such as the German Drought Monitor. Furthermore, such an approach, which explicitly captures the complexity of the underlying reaction mechanism rather than relying on one major determinant, generally appears to be more suitable for the projection of climate impacts, since GCM explicitly capture

the dynamics of several hydro-meteorological variables (Crane-Droesch, 2018). However, further research is needed to better take into account small-scale events such as hail and thunderstorms and to better reflect region-specific differences in growth periods. The compounding effects of interacting characteristics also need to be studied in more detail and should be clarified using appropriate methods. In addition, it is important for seasonal forecasting to improve the ability to predict events outside the sample. Here, e.g. the use of deep learning instead of classical machine learning could help to further improve predictive

capabilities. Annual patterns not covered by this approach could also be more explicitly captured. Similarly, a sensitivity analysis of the expert thresholds used to define the extreme values could help to improve the model.

*Code and data availability.* The input data and the script for processing the data and for analysis are available on the following UFZ repository: https://git.ufz.de/damage-functions/rf-winterwheat.



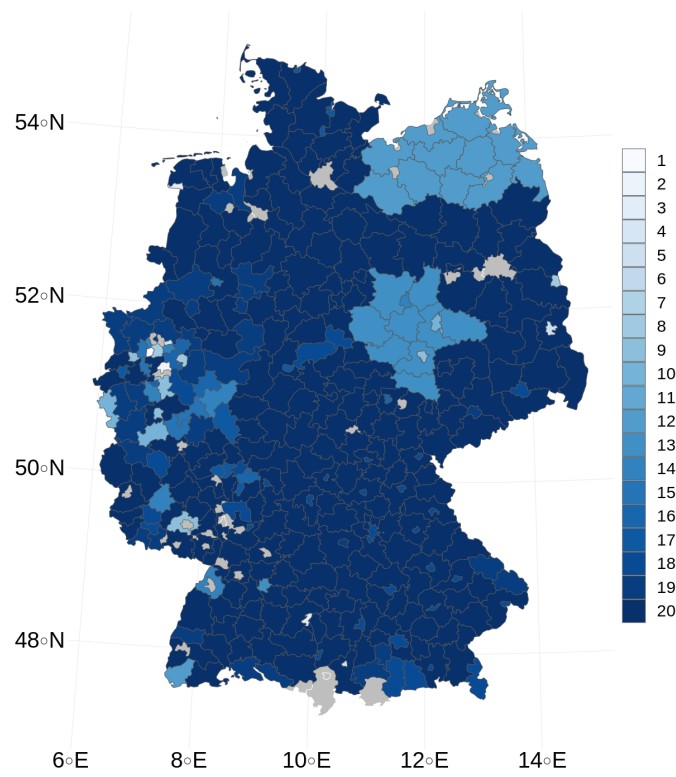

**Figure A1.** Map showing the number of winter wheat yield observations available for each counties for the period 1999 - 2018.

## Appendix A

The appendix includes additional information on data, cluster validation approaches, as well as variable importance plots and
accumulated local effects plots for the best combination of cluster algorithm, cluster size and SMI for a particular soil depth,
i.e. Partitioning Around Medoids (PAM) with 8 clusters and soil moisture for the uppermost 25 cm, not shown in the main text.

### A1   Map of available yield observations for each county

We use a spatio-temporal data set containing 410 counties and 20 years. All counties with less than ten years of reported yields
are excluded from the analysis (figure A1). There were 350 remaining counties in total.





## A2 Correlation plot of the soil moisture index for the entire root zone for all months of the season of winter wheat.

Figure A2 shows the correlation of soil moisture indices for total root zone depth for the season of winter wheat in Germany from October to August. This correlation shows the persistence of soil moisture and the smoother distribution resulting from it compared to meteorological variables. The Pearson correlation coefficient between the neighbouring SMIa is between 0.62

and 0.95. For the second order neighbours it is still between 0.42 and 0.88. In general, the largest correlation coefficients are found for the first half of the season. For this reason, within the Random Forests, we consider only the months of October, January, April and July.

## A3 Cluster Validation

Here, we use internal validation measures to assess the quality of the clustering, which employ only the data set and the

clustering partition for the assessment Subbaswamy (1977). The specified measures are connectivity, silhouette width, and Dunn index (see Figure A3). Connectivity refers to the degree of connectivity of the clusters (Handl et al., 2005). It has a value between 0 and infinity and should be minimized. Both the silhouette width and the Dunn index represent linear combinations of compactness and separation of the clusters. The Dunn index has a value between 0 and infinity and should be maximized (Dunn, 1974). The silhouette width ranges between -1 and 1 and well clustered observations have a value close to 1 (Rousseeuw,

1987). The connectivity mainly indicates the use of small number of clusters, Dunn, at the other end, rather large number. Silhoutte Width, on contrast, prefers a rather small number of clusters. In all three approaches the HIERARCHICAL algorithm is preferred. As a consequence of this ambiguity, we decided to evaluate the cluster algorithm and the number of clusters by the R-square outside the sample, which is generated for each cluster and the number of cluster combinations for the separate soil moisture configuration.

## 360 A4 Accumulated local effects plots

The ALE plots for the best combination of cluster algorithm, size, and SMI for the corresponding eight clusters are shown in Figure A4. The spatial arrangement of the clusters can be seen in Fig.2 of the main text. The six most important features are shown for each cluster. As shown above, this ranking of importance is associated with a large uncertainty (not shown). These ALE plots give a more detailed description of the damage mechanism for subregions in Germany. However, they are more

erratic than those shown before, which could indicate an over-fit.

## A5 Variable importance plots

Here, importance is defined as the factor by which the model's mean average error (mae), a measure of model performance, changes when the feature is shuffled Molnar (2020). To overcome the randomness added by this shuffling, the permutation is repeated 50 times and the results are averaged. Hence, the results show a large variability, especially in the most important

features (figure A5). Moreover, the less data are available, the greater is the variability of the results. Cluster 2 has the smallest number of counties compared to cluster 1 and the non-cluster approach. As figure A5a shows for a non-cluster approach ten

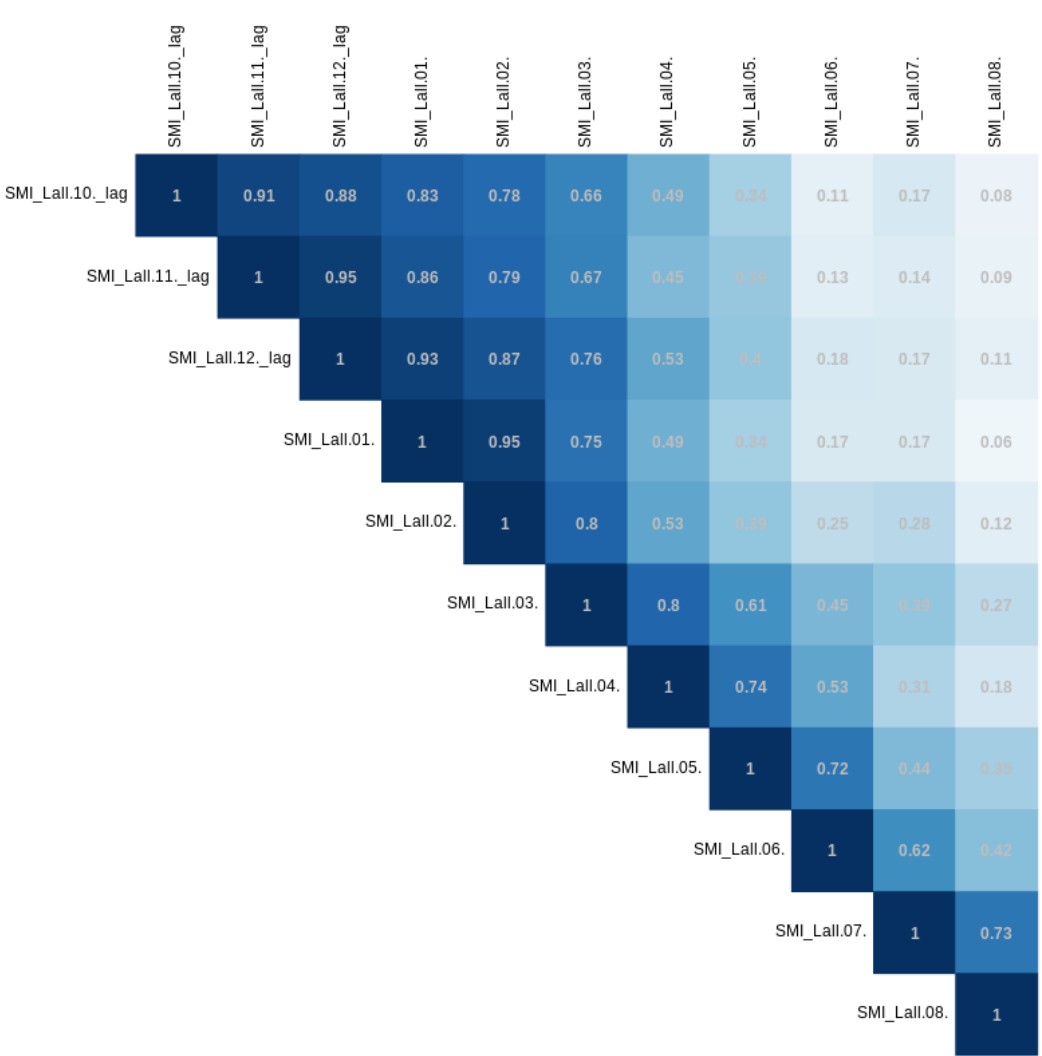

**Figure A2.** Correlation plot (Pearson correlation coefficient) of the soil moisture index for the entire root zone for all months of the season of winter wheat.





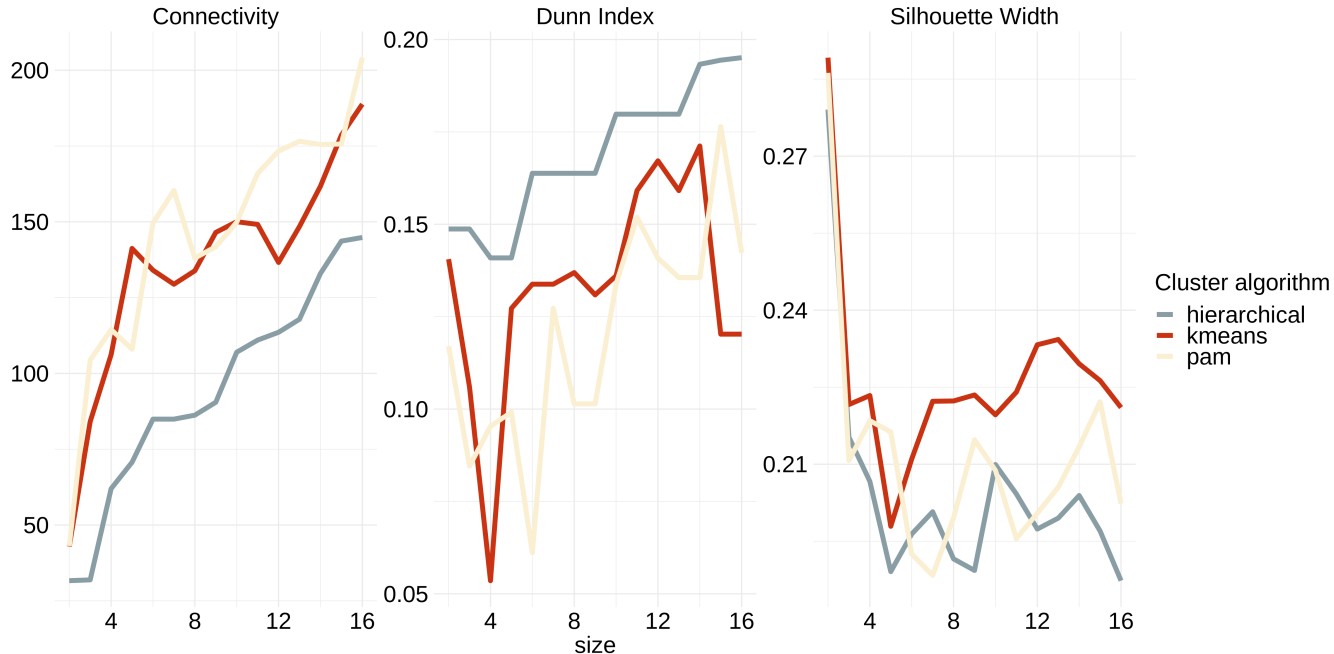

**Figure A3.** Internal validation measures for clusters with different sizes between 2 and 16. The measures depicted are Connectivity, Dunn Index, and Silhouette Width.

out of the twelve most important variables are soil moisture in the uppermost 25cm during different times within the growing season and March being the most important month. The most important meteorological variable is Heat for August. Cluster 1 represents almost the same variables as those found in the non-cluster configuration (figure A5b). Only PS5 is considered

instead of PS4. However, the order of the variables changes. In particular the most important meteorological variable Heat8 is less important in cluster 1. Overall, SMI of March is still the most important variable. Also, the two lagged soil moisture variables gain relevance. For one of those two, i.e. SMI in November(Heat8), the largest variability can be found. For cluster 2, a new picture evolves as four different variables are considered here (figure A5c). In particular lagged soil moisture values of the year before are not considered as well as February soil moisture. Also, PS5 is not represented in the data there. Instead,

precipitation scarcity of April, July, and August is considered now. This indicates, that the meteorological variables are more important in the regions considered here. Also, the late spring and summer seasons are more pronounced as precipitation scarcity in April as well as soil moisture from May to July are amongst the most important variables. However, soil moisture in March is still the most relevant variable. In general soil moisture supports the performance of the model for all three considerations the most. This is particular true for the non-cluster approach and cluster 1 as in cluster 2 more meteorological

variables are critical. The most important variable for all three cluster considerations is the same. The only meteorological variable listed for all three clusters is Heat8. It can be observed that the non-cluster approach particularly reflects cluster 1 whereas cluster 2 is underrepresented.



**Figure A4.** Accumulated local effects (ALE) plots for the best combination of cluster algorithm, cluster size and SMI, i.e. PAM with 8 clusters and soil moisture for the uppermost 25 cm. For each cluster ((a) - (h)) the six ALE plots with the highest feature importance are shown. The importance ranking is established with 50 reputations. We have chosen a rather large interval size of 50 to estimate the ALE plots, which allows us to reveal the true complexity of the model at the expense of shakiness. Therefore a nonlinear smoothing function (LOESS - locally estimated scatterplot smoothing) is added in blue (with confidence interval in grey).





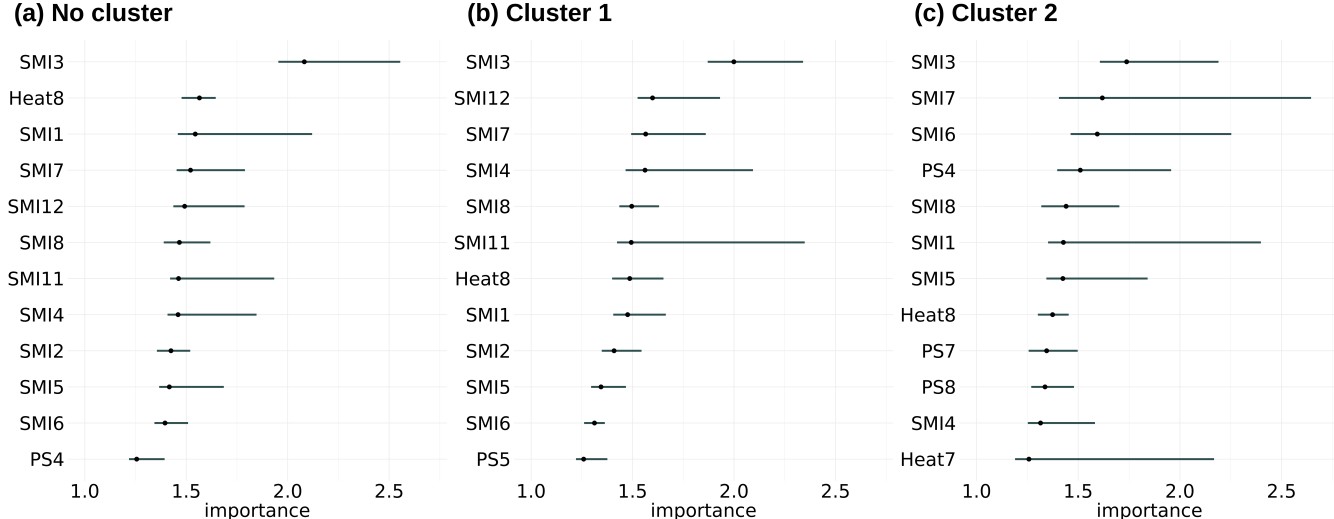

**Figure A5.** Variable importance of the twelve most important features for no cluster (a), cluster 1 (b), and cluster 2 (c). SMI represents the soil moisture index for the uppermost 25 cm of the soil column, PS stands for days without rain in a given month and Heat for days with a maximum temperature of more than 30 degrees. The number between the two points indicates the month, refers to the year before. For example, Frost10 represents black frost in October.

*Author contributions.* A.M. and S.T. prepared the historical meteorological data. A.M. applied the hydro-meteorological simulations. S.T. was responsible for the spatial processing of the data. M.P. developed the research idea, prepared the data and developed the statistical crop model. M.P. and A.M. analysed the results. M.P. composed the text. M.P., S.T., L.S., B.H. and A.M. contributed to interpreting results.

*Competing interests.* The authors declare that they have no conflict of interest.

*Acknowledgements.* This work was partially supported by funds through the projects CLIMALERT (project no.ERA4CS/0005/2016). The yield data are provided by the Regional Statistics Germany (https://www.regionalstatistik.de/). The input data and the script for processing the data and for analysis are available on the following UFZ repository: https://git.ufz.de/damage-functions/rf-winterwheat.



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
