# Peer review of "Machine learning methods to assess the effects of a non-linear damage spectrum taking into account soil moisture on winter wheat yields in Germany"

_Hydrology and Earth System Sciences, 2021_

## Referee Comment (RC1)

**Review on Peichl et al. 2021**

**Summary**

The authors present an assessment on the relationship of wheat yield anomalies to hydro-meteorological features using cluster analysis and random forest together with accumulated local effects. They achieve good model performance and additionally present an in-depth investigation of the most important driver variables, which adds additional value to the article. They highlight that their results serve well for the regional identification of harmul seasonal hydro-meteorological conditions. Furthermore, they state the potential usage of their research e.g. for the identification of harmful features and their related thresholds and also carefully underline the limitations regarding out-of-sample predictions, underestimations of extremes and missing inclusion of interaction effects in the model. I suggest that the manuscript should be accepted with moderate revision.

**Major comments**

The fact that the clusters are identical with the borders of the federated states of Germany (or groups of them) without a single exception (as far as I can see) is not discussed and does not seem obvious to me. The authors should provide an explanation for this, e.g. stating which of the included variables in the cluster analysis are so strikingly different between the federated states that it leads to a perfect match of clusters and federated states (while any environmental / climatic differences do not matter in the clustering). It seems plausible that differences between eastern and western Germany due to the different political systems in place in the past can have a significant influence, but I wonder why this should hold true at the administrative level of federated states. The variables included in the cluster analysis are average yield and monthly averages and daily observations of the meteorological data for the entire year and SMI for both the upper layer and the entire soil column, right? I cannot see how these variables should primarily be superposed by the shapes of the federated states of Germany, even though I understand your argument that average yield is connected to farm size (which differs strongly between eastern and western Germany).

**Minor comments**

Title: The paper is only about random forest, so you could consider using this term here instead of the more general term "machine learning".

l. 11: "R-sqare**d**" or preferably "$R^2$". Make this consistent throughout the paper.

l. 16: I assume you mean "crop yield variations".

l..79: Is the trend statistically significant?

Figure 1, 2, 5, A1: There are various grey counties, I assume mostly cities and/or counties without nonirrigated arable land (Peichl et al. 2018). If all these counties are the ones with 0 years of data, this should be clearly visible and stated in Fig. A1.

Figure 1 caption: Consider specifically naming the exceptional years 2003, 2014 and 2018 here.

Table 1: It should say "**max**. T>3°C" for alternating frost according to Gömann et al. 2015

Table 1: The presented variables in Table 1 seem plausible, but I wonder how excactly you came up with them? You sometimes depart from the months suggested by Gömann et al. 2015, e.g. they state the usage of Jan-Apr for Alternating Frost, while you also included May.

Table 1: It would be beneficial to have all predictor variables for the random forest in one overview table, so consider including SMI here, too.

l. 100: "each grid **point** depends"

l. 101-102: You compare your soil moisture index to simulations, but the soil moisture index used here is also simulated, so I do not understand this sentence fully. Could you elaborate please?

l. 113, 350, 368: adjust citation brackets

l. 113: I understand the line of argumentation of the authors, that soil moisture is a slow-responding variable with a long memory, but relating this to autocorrelation per se might be a bit misleading here, because temperature is of course also auto-correlated. So consider writing something like "The autocorrelation / long-term temporal persistence of soil moisture is comparatively high in comparison to temperature".

l. 116: State here also specifically that on the other hand for SMI all months of the growing season were used (only SMIa is reduced to four months).

l. 119: My understanding is that masked grid cells are excluded, but you mean the opposite that you kept only non-irrigated agricultural land, right?

l. 145: Did you run the random forest model for each cluster size from 2 to 16 for each algorithm as you did for your standard internal validation?

l. 147-49: I wonder whether the statements on the data inlcuded in the cluster analysis should be rather mentioned adjacent to the data included in the random forest model in section 2 to have the complete overview at once.

Table 2: While SMIa stands for the entire soil column, usage of the term SMI is ambigous. You use it to refer to the uppermost 25 cm (e.g. l.111), but also as a general term for both depths (e.g. l. 148, Table 2). Consider using separate terms.

Figure 2: You use dark grey for missing data and light grey for cluster 2. Consider taking another color for cluster 2 to make it better distinguishable.

Figure 2 and A1: coordinate degree sign as superscript

l. 189: You mention structural differences between western and eastern Germany. Consider mentionening briefly here the large differences in farm sizes due to different political systems in place, as an international audience might not be aware of this.

Figure 3: Indicate that the regression line is shown in bold black in the figure description. I assume the ellipses are point densities? Please specify this.

l. 227: I assume this reduction is to be compared with an average month, right?: "for each month with an SMI value lower than 0.125 **compared to an average month.**"

Figure 4: Is there a reason why there are many more red dots for the SMI compared to Heat and PS in a given subpanel?

Figure 4: You state for Fig. A4 and in the text section describing Fig. A5 that the average of 50 repititions is taken. Is this also the case here?

Figure 4 and A4: You chose an interval size of 100, whereas for Figure A4 it is 50. Why did you choose different interval sizes?

236: SMI **year-round**. "entire" could be confused with the entire soil column.

l. 239: "**led** to large losses"

l. 241: "but **need** to be"

l. 243: I assume "crop **yield** potentials" might be more precise.

l. 245: "water govern**s**"

l. 246: delete "this"

l. 266: "**eleventh** day"

l. 268: Why especially East Germany? It is valid for non-cluster and cluster 1, too, is it not?

l. 274: delete "years" in front of "in-sample"

l. 274: It did not directly become clear to me that 2019 is an out-of-sample year at the first read. I suggest explicitly naming it an out-of-sample year in this context (as you also do below).

l.286: "in **some of** the easternmost"

l. 288: "the **highest** negative"

l. 314: You use the terms East / eastern Germany and northeastern Germany. If all these mean the same region, consider using only one term or are you specifically referring only to cluster 7 with northeastern Germany?

l. 324: GCM: Abbreviations should be written out at first occurrence.

Code and data availability: The scripts seem not available online yet.

l. 355: "**a** small number" "**a** rather large number"

l. 367: Do you mean "mean **absolute** error"?

l. 376/378: Do you refer to SMI11 and SMI12 as "lagged"? Simply because they belong to another calendar year does not make them "lagged" variables in my opinion, so this term might be inappropriate here.

l. 377: Is SMI11 or Heat8 meant here?

Figure A4 caption: "50 **repetitions**"

Figure A5 caption: The caption should specify that the range of each variable stems from 50 (or 100?) repititions.

---

## Author Comment (AC1)

**Rebuttal Letter for the Reviews on 'Machine learning methods to assess the effects of a non-linear damage spectrum taking into account soil moisture on winter wheat yields in Germany'**

Michael Peichl[1*], Stephan Thober[1], Luis Samaniego[1], Bernd Hansjuergens[2], and Andreas Marx[1**]

[1]UFZ-Helmholtz Centre for Environmental Research, Department Computational Hydrosystems, Permoserstrasse 15, D-04318 Leipzig, Germany
[2]UFZ-Helmholtz Centre for Environmental Research, Department Economics, Permoserstrasse 15, D-04318 Leipzig, Germany
*michael.peichl@ufz.de
**andreas.marx@ufz.de

We thank the reviewers for their detailed comments, which helped us to improve our manuscript substantially. In particular, the main comment from reviewer 1 was was very helpful because the corresponding analysis of why the clusters are identical to the state boundaries helped us identify an issue within the data provided to the clustering algorithms that was not valid. Unfortunately, this analysis is the basis of the statistical analysis and therefore has profound implications for all of the results presented in this manuscript.

**Reviewer 1**

**Summary**

*The authors present an assessment on the relationship of wheat yield anomalies to hydrometeorological features using cluster analysis and random forest together with accumulated local effects. They achieve good model performance and additionally present an in-depth investigation of the most important driver variables, which adds additional value to the article. They highlight that their results serve well for the regional identification of harmul seasonal hydro-meteorological conditions. Furthermore, they state the potential usage of their research e.g. for the identification of harmful features and their related thresholds and also carefully underline the limitations regarding out-of sample predictions, underestimations of extremes and missing inclusion of interaction effects in the model. I suggest that the manuscript should be accepted with moderate revision.*

**Dear Reviewer, thank you for your detailed summary and suggestions for accepting the manuscript with moderate revisions. We have responded to the major comments and the minor comments throughout the manuscript. Please find our responses below.**

**Major comments**

30  *The fact that the clusters are identical with the borders of the federated states of Germany (or groups of them) without a single exception (as far as I can see) is not discussed and does not seem obvious to me. The authors should provide an explanation for this, e.g. stating which of the included variables in the cluster analysis are so strikingly different between the federated states that it leads to a perfect match of clusters and federated states (while any environmental / climatic differences do not matter in the clustering). It seems plausible that differences between eastern and western Germany due to the different political systems*

35  *in place in the past can have a significant influence, but I wonder why this should hold true at the administrative level of federated states. The variables included in the cluster analysis are average yield and monthly averages and daily observations of the meteorological data for the entire year and SMI for both the upper layer and the entire soil column, right? I cannot see how these variables should primarily be superposed by the shapes of the federated states of Germany, even though I understand your argument that average yield is connected to farm size (which differs strongly between eastern and western Germany).* **We**

40  **sincerely thank you for this comment, as the corresponding thorough review of the cluster approach helped us identify a problem within the data provided for the cluster algorithms. Unfortunately, the data contained a variable that included state information, which was an artifact from the previous processing of the data. This data is not scaled and thus dominates the clustering results. For this reason, the resulting areas correspond to the state boundaries. We sincerely apologize for this error. It unfortunately affects all subsequent results as well.**

45  **Minor comments**

*[l. 11] "R-sqared" or preferably "$R^2$". Make this consistent throughout the paper.*

**Thank you for bringing attention to this unsightly habit. We use now "$R^2$" troughout the text.**

*[l. 16] "I assume you mean "crop yield variations".*

**Yes, we do.**

50  *[l. 79] Is the trend statistically significant?.*

**Yes, the trends in all three configurations (loess is defined as natural splines with 4 degrees of freedom) are significant. An ANOVA test shows, that the loess model with non-linear trend is significantly better than the linear model.**

*[Figure 1, 2, 5, A1] There are various grey counties, I assume mostly cities and/or counties without nonirrigated arable land (Peichl et al. 2018). If all these counties are the ones with 0 years of data, this should be clearly visible and stated in Fig. A1.*

55  **Thank you, we adapted the Fig. A1 and the description accordingly.**

*[Figure 1 caption] Consider specifically naming the exceptional years 2003, 2014 and 2018 here.*

**Thanks, the years are now named.**

*[Table 1] It should say "max. T>3°C" for alternating frost according to Gömann et al. 2015*

**Indeed, that is how we defined it. Thank you for pointing out this error to us.**

60  *[Table 1] The presented variables in Table 1 seem plausible, but I wonder how excactly you came up with them? You sometimes depart from the months suggested by Gömann et al. 2015, e.g. they state the usage of Jan-Apr for Alternating Frost, while you*

*also included May.*

**Thank you for drawing attention to this discrepancy. This research project also incorporated expert knowledge from interviews with farmers, which suggested the need to extend these time periods. Because the machine learning algorithm**
65 **used here inherently performs feature selection, we therefore extended the time periods suggested by Gömann et al. (2015) rather than restricting them.**

*[Table 1] It would be beneficial to have all predictor variables for the random forest in one overview table, so consider including SMI here, too.*

**For a better overview, all variables are now included in Table 1.**

70 *[l. 100] "each grid point depends"*

**Thanks, we use cells instead of points to improve clarity here.**

*[l. l. 101-102] You compare your soil moisture index to simulations, but the soil moisture index used here is also simulated, so I do not understand this sentence fully. Could you elaborate please?*

**Soil moisture is presented here as an index because an index configuration supports the reduction of systematic errors**
75 **of data that are simulated as well as spatially processed, such as in the present study (Auffhammer et al., 2013; Lobell, 2013)."**

*[l. 113, 350, 368] adjust citation brackets*

**Thank you for pointing this out.**

*[l. 113:] I understand the line of argumentation of the authors, that soil moisture is a slow-responding variable with a long*
80 *memory, but relating this to autocorrelation per se might be a bit misleading here, because temperature is of course also auto-correlated. So consider writing something like "The autocorrelation / long-term temporal persistence of soil moisture is comparatively high in comparison to temperature".*

**Thank you, we rewrote the entire passage to clarify the points you made. Cumulative measures are widely used for meteorological variables as for instance growing degree days and killing degree days (Schlenker and Roberts, 2009) .**

85 *[l .116] State here also specifically that on the other hand for SMI all months of the growing season were used (only SMIa is reduced to four months).*

**This is elaborated here and shown in Table 1 now.**

*[l. 119] My understanding is that masked grid cells are excluded, but you mean the opposite that you kept only non-irrigated agricultural land, right?*

90 **Grid cells that are not non-irrigated agricultural land are excluded and the remaining cells are used to calculate the respective county average.**

*[l. 145] Did you run the random forest model for each cluster size from 2 to 16 for each algorithm as you did for your standard internal validation?*

**Yes, we ran it for all possible combinations of cluster size (2-16) and clustering algorithms.**

95 *[l. 147-49] I wonder whether the statements on the data inlcuded in the cluster analysis should be rather mentioned adjacent to the data included in the random forest model in section 2 to have the complete overview at once.*

**We placed the data information here to avoid confusion with the data used in the Random Forests.**

*[Table 2] While SMIa stands for the entire soil column, usage of the term SMI is ambigous. You use it to refer to the uppermost 25 cm (e.g. l.111), but also as a general term for both depths (e.g. l. 148, Table 2). Consider using separate terms*

100 **We now use the abbreviation SMI only for the top 25 cm and when it is a general term we do not use the abbreviation SMI but write soil moisture index.**

*[Figure 2] You use dark grey for missing data and light grey for cluster 2. Consider taking another color for cluster 2 to make it better distinguishable.*

**The figure is now revised and the gray areas should now be easier to see.**

105 *[Figure 2 and A1:] coordinate degree sign as superscript*

**Thank you, we have not noticed this yet as it happened when converting the respective files.**

*[l. 189] You mention structural differences between western and eastern Germany. Consider mentionening briefly here the large differences in farm sizes due to different political systems in place, as an international audience might not be aware of this.*

110 **Thank you, we added this information backed by data from the Agrarstrukturerhebung in 2016.**

*[Figure 3] Indicate that the regression line is shown in bold black in the figure description. I assume the ellipses are point densities? Please specify this.*

**We added those information in Figure 3.**

*[l. 227] I assume this reduction is to be compared with an average month, right?: "for each month with an SMI value lower*
115 *than 0.125 compared to an average month."*

**By definition, an SMI of 0.5 indicates the 50 percent quantile of the empirical distribution of water content for that county and thus serves as a benchmark for the average month (assuming that the empirical distribution is symmetric). The value of 0.125 is taken here as representative of a threshold at which yield falls rapidly relative to average yield expectations. We have modified this section to make the interpretation clearer.**

120 *[Figure 4] Is there a reason why there are many more red dots for the SMI compared to Heat and PS in a given subpanel?*

**Meteorological features are discrete values (counting days above or below a predefined threshold), while SMI is a continuous feature between 0 and 1.**

*[Figure 4] You state for Fig. A4 and in the text section describing Fig. A5 that the average of 50 repitions is taken. Is this also the case here?*

125 **Thank you, the 50 in Fig. A5 refers to the number of repetitions of the permutation to define the variable importance. Instead of using the phrase interval size, we now use grid size in the contest of ALE-plots to avoid further confusion. The plot is based on the feature importance assessment with 50 repetitions and then the ALE plots are shown for the most important features. These plots are based on a grid size of 100.**

*[Figure 4 and A4] You chose an interval size of 100, whereas for Figure A4 it is 50. Why did you choose different interval sizes?*

**The grid size in Figure A4 of 50 was an artifact of previous procedures. The ALE plots shown here are also based on a grid size of 100.**

*[236] SMI year-round. "entire" could be confused with the entire soil column.*

**Here, "entire" referred to the entire range of the SMI in March. This has now been corrected.**

*[l. 239] "led to large losses"*

**Thank you, changed accordingly.**

*[l. 241] "but need to be"*

**Thank you, we changed it accordingly.**

*[l. 243] I assume "crop yield potentials" might be more precise.*

**Thank you, to include "yield" is indeed more accurate.**

*[l. 245] "water governs"*

**The structure of the entire sentence was odd so we changed it: "The observation that the absence of water governs crop production in eastern Germany is in alignment with recent studies."**

*[l. 246] delete "this"*

**See reply above.**

*[l. 266] "eleventh day"*

**We changed it accordingly.**

*[l. 268] Why especially East Germany? It is valid for non-cluster and cluster 1, too, is it not?*

**Thank you for bringing this ambiguity to our attention. We have changed the entire paragraph there: "Our approach, which explicitly controls for plant water supply through soil moisture, generally shows (for the no cluster approach as well as cluster 1 and cluster 2) more negative effects in terms of water deficit compared to heat. Especially for eastern Germany (cluster 2), the water deficit in the upper 25 cm of the soil plays a prominent role in late summer and spring."**

*[l. 274] delete "years" in front of "in-sample"*

**We have deleted years.**

*[l. 274] It did not directly become clear to me that 2019 is an out-of-sample year at the first read. I suggest explicitly naming it an out-of-sample year in this context (as you also do below).*

**We added this information to clarify in the beginning of this paragraph that 2019 was not used in the training set.**

*[l.286] "in some of the easternmost"*

**Thank you, we changed it accordingly.**

*[l. 288] "the highest negative"*

**Thank you, we adopted this.**

*[l. 314] You use the terms East / eastern Germany and northeastern Germany. If all these mean the same region, consider using only one term or are you specifically referring only to cluster 7 with northeastern Germany?*

**Thank you, we changed the term here to eastern Germany. For the rest of the text we use eastern Germany when the** 165 **spatial scope is the focus, and East Germany when the political system and the history of the region is of particular interest.**

*[l. 324] GCM: Abbreviations should be written out at first occurrence.*

**Thank you, we introduced global climate models.**

*[l. 355] "a small number" "a rather large number"*

170 **Changed accordingly, thank you.**

*[l. 367] Do you mean "mean absolute error"?*

**Yes, indeed, the mean absolute error is meant.**

*[l. 376/378] Do you refer to SMI11 and SMI12 as "lagged"? Simply because they belong to another calendar year does not make them "lagged" variables in my opinion, so this term might be inappropriate here.*

175 **Thank you, to avoid any confusion when it comes to those terms we instead refer to the previous years.**

*[l. 377] Is SMI11 or Heat8 meant here?*

**SMI11 of course.**

*[Figure A4 caption] "50 repetitions"*

**Thank you, we adopted this.**

180 *[Figure A5 caption] The caption should specify that the range of each variable stems from 50 (or 100?) repititions.*

**Thank you, we included this information now.**

**References**

Auffhammer, M., Hsiang, S., Schlenker, W., and Sobel, A.: Using Weather Data and Climate Model Output in Economic Analyses of Climate Change, Tech. Rep. 2, National Bureau of Economic Research, Cambridge, MA, https://doi.org/10.3386/w19087, http://www.nber.org/papers/w19087.pdf, 2013.

Gömann, H., Bender, A., Bolte, A., Dirksmeyer, W., Englert, H., Feil, J., Frühauf, C., Hauschild, M., Krengel, S., Lilienthal, H., Löpmeier, F., Müller, J., Mußhoff, O., Natkhin, M., Offermann, F., Seidel, P., Schmidt, M., Seintsch, B., Steidl, J., Strohm, K., and Zimmer, Y.: Agrarrelevante Extremwetterlagen und Möglichkeiten von Risikomanegementsystemen: Studie im Auftrag des Bundeministeriums für Ernährung und Landwirtschaft (BMEL); Abschlussbericht: Stand 03.06.2015., Tech. rep., Johann Heinrich von Thünen-Institut, https://doi.org/10.3220/REP1434012425000, 2015.

Lobell, D. B.: Errors in climate datasets and their effects on statistical crop models, Agricultural and Forest Meteorology, 170, 58–66, https://doi.org/10.1016/j.agrformet.2012.05.013, https://linkinghub.elsevier.com/retrieve/pii/S0168192312001906, 2013.

Schlenker, W. and Roberts, M. J.: Nonlinear temperature effects indicate severe damages to U.S. crop yields under climate change, Proceedings of the National Academy of Sciences, 106, 15 594–15 598, https://doi.org/10.1073/pnas.0906865106, http://www.pnas.org/cgi/doi/10.1073/pnas.0906865106, 2009.

---

## Author Comment (AC2)

**Rebuttal Letter for the Reviews on 'Machine learning methods to assess the effects of a non-linear damage spectrum taking into account soil moisture on winter wheat yields in Germany'**

Michael Peichl[1*], Stephan Thober[1], Luis Samaniego[1], Bernd Hansjuergens[2], and Andreas Marx[1**]

[1]UFZ-Helmholtz Centre for Environmental Research, Department Computational Hydrosystems, Permoserstrasse 15, D-04318 Leipzig, Germany
[2]UFZ-Helmholtz Centre for Environmental Research, Department Economics, Permoserstrasse 15, D-04318 Leipzig, Germany
[*]michael.peichl@ufz.de
[**]andreas.marx@ufz.de

We thank the reviewers for their detailed comments, which helped us to improve our manuscript substantially. In particular, the main comment from reviewer 1 was was very helpful because the corresponding analysis of why the clusters are identical to the state boundaries helped us identify an issue within the data provided to the clustering algorithms that was not valid. Unfortunately, this analysis is the basis of the statistical analysis and therefore has profound implications for all of the results presented in this manuscript.

**Reviewer 2**

**General comments**

*This paper presents the modelling of non-linear effects of meteorological divers and soilmoisture on winter wheat yield variability. A random forest procedure models the nonlinear relationships. The model is applied on subregions of Germany, obtained with a clustering procedure. A comparison with the model trained over the whole country emphasizes the relevance of the clustering. The authors highlight the importance of soil moisture as a relevant explicative variable, more relevant than heat. The manuscript is well written. The description of the results is clear and supported by existing literature. This paper deserves publication after minor modifications/addition of complementary information.*

**Dear Reviewer, thank you for your detailed summary and suggestions for accepting the manuscript with minor modifications/addition of complementary information. We have responded to the specific comments and technical corrections throughout the manuscript. Please find our responses below.**

**Specific comments**

*[Table 1] The description of variables is generally self-explanatory, except maybe "alternative frost". Does it refer to the*
30  *number of consecutive pairs of days with min T<-3 and then minT>3?*

**Thank you for drawing attention to this need for clarification. First, the minT > 3 should be declared as maxT > 3. Also, it takes into account days when both conditions apply, i.e. a day with, for example, minimum temperatures below -3 degrees at night and then during the day with degrees higher than -3 maximum temperature.**

*Is there a correlation between SMI and SMIa, and can this have an impact on the quality of the RF model? Same question for*
35  *correlation between SMI(a) and indicators such as Heat, Heavy rain, precipitation scarcity.*

**There is a correlation between SMI and SMIa as well as with all meteorological variables. However, it is generally understood that this does not affect the predictive capacity of Random Forests - and this is what we assume is meant by model quality here. Nevertheless, it does affect the interpretability of the model. In our case, this is in particular the case for feature importance (if the are multiple features that contain the same information this as an impact of the**
40  **order of partitioning the data). Since we are using accumulated local effects, purged of any correlation, this should be less affected by correlation issues such as multicollinearity.**

*[l.118/119] SMI is masked for non-irrigated agricultural land, but are these areas also discarded in yield data?*

**As the yield information are only available on administrative district level this masking was not possible. When it comes to the factor irrigation, we consider this neglectable as only about 5 percent of the agricultural area is possibly irrigated**
45  **with an focus on crops like potatoes (https://www.destatis.de/DE/Themen/Branchen-Unternehmen/Landwirtschaft-Forstwirtschaft Fischerei/Produktionsmethoden/Tabellen/bewaesserungsmoeglichkeiten.html)**

*[l.138] How can one interpret quickly subregions within Germany obtained with clustering? Are they areas where yield is of the same order of magnitude and also monthly and daily meteorological are also similar?*

**A more detailed description can be found in our response to the main comment of reviewer 1. Unfortunately, the cluster**
50  **algorithm is provided with invalid data regarding the federal states that impact the cluster formation.**

*[Figure 2] Please specify in the caption or the legend that the numbers in rectangles are referring to the Rsquare obtain from the RF procedure(?).*

**The entire figure now has been revised. Now, we state in the caption that the number indicate the respective test R-squared.**

55  *[Figure 2a] Is it by chance that except cluster number 8, all clusters are simply connected and almost convex? What could explain this very smooth partition?*

**Please see reply above and to reviewer number one.**

*[l.186] Would it a better option to use PAM(3) (with only SMIa) than PAM(2) (with both SMI)? (To get rid of potential correlation problems between SMI and SMIa).*

60  **We apologize for the confusion caused by not further clarifying that we are relying only on the top 25 cm in this setting.**

**To fix this problem, we have added this sentence to the paragraph before it: "Since the data for the entire soil column do not appear to provide any additional information for the model, we rely only on the top 25 cm for further analysis."**

*[l.202] What would be a solution to avoid this overfitting of the model?*

**In this case, we compare the ALE plots of RFs fitted with either the PAM8 or the PAM2 setting. Thus, the former**
65 **is clustered into eight subregions. This means that each RF model is trained on a smaller sample size such as in the PAM2 context or when no cluster is used. Consistently, this is also the model and corresponding sample in each case on which the ALE plots for that subregion are based. Because of the less smooth functional relationships shown for this configuration, we assume that the models are overfitted to this small sample size. One possible solution would be to fit to larger samples. This is done here by relying instead on PAM2, i.e., the setting that considers only two clusters and**
70 **therefore has a higher sample size in each cluster.**

*[l.202] "The effects shown here are additive as they are cleared off the correlation to other features". I don't fully understand this sentence. Could you be more specific?*

**When features interact with other features in a prediction model, the prediction cannot be expressed as the sum of the feature effects, because the effect of one feature depends on the value of the other feature. Because of this compounding,**
75 **the features are correlated to each other. In linear models this is commonly expressed by multiplicative expressions. As the ALE plots are purged of this correlation, we can rely on the sum of this features to derive the overall effect. This is similar to a linear model with only additive terms. We added some clarifications: "The feature effects shown here can be interpreted as additive because they are purged of correlation to other features. For example, the combined effect of soil moisture in June and July is the sum of SMI6 and SMI7."**

80 *[Figure 4] Do the black and white bars in x-axis represent the distribution of the explicative variable?*

**Yes, the black bars show the distribution of the respective features. White bars a results of discrete features.**

*[Figure 4., caption] It is not clear to me what the interval size of 100 refers to.*

**ALE plots predict the effect of an explanatory variable across their realisations, taking into account only a subset of the sample with observed values adjacent to the respective realisation. The size of this subset of the sample is defined by the**
85 **grid size. The larger the grid size, the smaller the subset and the less smooth the visualization of the average marginal effects.**

*[l240] Would it be possible to add simple interactions in the model? (multiplication of 2 variables?)*

**In general, it would be possible to add these interactions by simply converting the features themselves into interactions, e.g. by multiplication of two variables as suggested. However, due to the strongly nonlinear structure caused by the**
90 **underlying recursive partitioning, we consider such an approach not necessary for (large enough) random forests.**

*[l265-266] "In both clusters, heat in August, a period generally associated with ripening, has positive effects for each additional day and from day 11 onward negative effects" According to fig4, it looks like only for cluster 1, Heat8 has a negative effect from day 11 onward, not for cluster 2.*

**Thank you, we now distinguish between cluster 1 and cluster 2: "In both clusters, heat in August, a period generally**

95  associated with ripening, has positive effects for each additional day and negative effects after the eleventh (cluster 1) respective sixth (cluster 2) day."

*[l.291-295] Can the difficulties of out-of-sample prediction be interpreted as overfitting? Could it be Improved with longer time series (to have a larger number of configurations)?*

The validation criterion the random forests are based on are out-of-bag error estimates. With tree sizes large enough
100 the estimates converge to those found for leave-one-out cross-validation. This validation technique is more prone to overfitting compared to other methods. Potential solutions are the use of 1) an extended time-series, 2) the use of a different cross-validation technique with higher focus the variabilty of the prediction but less the bias in the training data, 3) the use of machine learning techniques that might be better suitable to extrapolated out-of-the sample compared to random forests. We clarified this in more detail in this paragraph and the conclusions.

105 **Technical corrections**

*[l.113] Citation in brackets*

**Thank you, the citations are now corrected.**

*[l.154] Missing bracket*

**Thank you, we included the missing bracket.**

110 *[l.202] Extra "the" in "The effects shown here are additive as the they are cleared"*

**Thank you, we deleted the extra "the"**

*[l.377] Is "(Heat8)" supposed to be in that sentence?*

**No, it is supposed to be SMI11. Thanks for bringing this to our attention.**

*[Caption figure A4] "50 repetitions"*

115 **Thank you, it is of course 50 repetitions.**

---

## Referee Report (RR1)

**Second review on Peichl et al. 2021**

The state-dependent clustering was removed in this version and all relevant figures have been updated accordingly. I therefore recommend the article for publication after minor revision.

**Major**

Table 2 and lines 180-181: In all 12 cases the test $R^2$ is higher than the train $R^2$. In my understanding, test performance is usually below training performance. Higher test performance can occur occasionally, but it seems odd that this is the case for all your settings. Do you have an explanation for this? How did you split the train and test data set? Is there e.g. any possibility of a bias due to the splitting? Did you try various ways of splitting the data?

Related to this: You mention overfitting as an explanation of underestimation of year 2019 in line 328-329. But your better test performance compared to training performance would not indicate overfitting. (However, the following argument in line 330-334 seems reasonable.)

**Minor**

L. 65: What are mean average effects? Mean and average seem synonymous.

Figure 1: Isn't it 2003 instead of 2004?

Table 2: If you explain the abbreviations T and P here, you should also explain SMI and SMIa for the sake of completeness.

L. 220-221: SMI4 (5th in non-cluster); SMI8 (6th in non-cluster)

Figure 4: Why is it not completely identical for "4a) no cluster" to the submitted version? SMI4 is now on fourth place instead of SMI12.

Figure 4: Mention in the caption that this is for PAM (4).

Figure 4, A5-A9: You mention a grid size of 50, however in your answer to the reviews you state 100 (lines 139-148 of your rebuttal letter).

L. 243: "A low drought signal of soil moisture is found for April (SMI4)." Odd phrasing, consider rephrasing.

L. 268: What about cluster 1 and 3? They also exhibit drought vulnerability. Maybe write "subregion such as cluster 2..."

L. 270 and 338: crop **yield** potential

Figure 5: Missing data in gray should be indicated in the caption.

Fig. A4-A6: What is CR12?

L. 405: It is mean **absolute** error, not average. This was already mentioned in the first revision.

**ALE Plots**

The ALE plots are confusing because the contain a variety of small mistakes and inaccuracies. Basically I think you want to show (as indicated in lines 414-418) in these plots the 3 best cluster algorithms for cluster 2, 4 and 6 for the two soil moisture index configurations "Soil moisture for uppermost 25cm" and "Soil moisture index for both uppermost 25cm and entire soil column" (one of the plots (Fig. 4) is in the main text), right?

L. 376-378: This text was not updated. It is no longer PAM (8).

Fig. A5 is not for PAM with 2 clusters as indicated in the main text and the caption, but identical to Fig. A6.

Figs. A5-A9 all say "for both cluster". In three of these figures there are more than two. Also the plural-s is missing.

First line of the caption: Instead of "SMI" call this "soil moisture index configuration" in all ALE plots as you do in Table 2.

Fig. A5 first line of the caption: Shouldn't this be HIERARCHICAL (2) instead of PAM (2)? According to Table 2 HIERARCHICAL (2) is the best algorithm cluster size 2 and soil moisture index configuration "soil moisture for the uppermost 25 cm." Also in Line 414-415 it is mentioned as HIERARCHICAL (2).

Fig. A7 second line of the caption: 2 clusters, not 6.

Figs. A5-A9 first sentence of the caption: This is a bit misleading (in all ALE plots). It should rather say something like: best combination of cluster algorithm (PAM) and the soil moisture index configuration "soil moisture for the uppermost 25 cm" for cluster size 2. Each plot shows a different cluster size, so it is not part of the best combination, but rather it is the best combination for a given cluster size.

**Typos**

L. 115: extra bracket

L: 233: bracket missing

L. 240: 4**d**

L. 249: **H**eat8 in upper case

L 247: bracket missing

---

## Referee Report (RR2)

**Machine learning methods to assess the effects of a non-linear damage spectrum taking into account soil moisture on winter wheat yields in Germany**

**General comments**

The authors apply a random forest procedure to explain observed yield anomaly in Germany thanks to meteorological predictors. The central result of this paper is the quantification of individual non-linear contributions of meteorological variables/indices and especially the important role of soil moisture. The ALE plots are valuable material in this study (I am not sure about the interpretation of confidence intervals though, see specific comment n.3). To my opinion, this paper deserves publication after minor revisions.

**Specific comments**

1. Fig 1b. You mention the identification of significant trends, did you perform some trend test or shift test?
2. l141. Is the clustering performed on raw yield directly, or anomalies? How should one interpret the clusters obtained: are the counties gathered in terms of yield magnitude or variability or occurrence of extremes?
3. Figure 4: the confidence intervals you obtain are related to the smoothing function, and not the robustness of the RF model itself. I am wondering to what extent it is possible to interpret it as an uncertainty of the variable effect (ex. l.238). To my understanding, this confidence interval tells us about the uncertainty of the smoothed curve, but not about the uncertainty of the local effect.
   (small remark: it is nice to specify the package for the ALE).
4. l. 405 Do I understand correctly that "the feature is shuffled" means that the variable, for which we want to compute the importance, is shuffled?. I.e., to get the importance of e.g. SMI3, the RF is re-run on the exact same data, except for SMI3, which is shuffled in time, and then the MAE is computed?

**Technical corrections**

1. l233. missing ")".
   l240 error in the reference to the figure
   l257 missing ")"

It seems like there are some confusions in the figure captions:
2. Caption figure 3 not "observed and predicted data for the *two* clusters" but "*four* clusters"?
3. Figure 5: you chose to conduct the analysis with PAM 4 clusters, so in Fig5 it is four and not eight subregions, right?
4. Figure A5 and A7: the captions don't seem to fit with the figures (clustering method and number of clusters).
5. l.378: aren't the results in the appendix about PAM 4 and 6, and hierarchical 2 and 6?

---

## Author Response (AR2)

**Rebuttal Letter for the Reviews on 'Machine learning methods to assess the effects of a non-linear damage spectrum taking into account soil moisture on winter wheat yields in Germany'**

5    Michael Peichl[1*], Stephan Thober[1], Luis Samaniego[1], Bernd Hansjuergens[2], and Andreas Marx[1**]

[1]UFZ-Helmholtz Centre for Environmental Research, Department Computational Hydrosystems, Permoserstrasse 15, D-04318 Leipzig, Germany
[2]UFZ-Helmholtz Centre for Environmental Research, Department Economics, Permoserstrasse 15, D-04318 Leipzig, Germany
10    [*]michael.peichl@ufz.de
     [**]andreas.marx@ufz.de

We thank the reviewers for their detailed comments, which helped us to improve our manuscript. Below are the responses corresponding to Reviewer 1's and Reviewer 2's comments.

**Reviewer 1**

15    **Summary**

*The state-dependent clustering was removed in this version and all relevant figures have been pdated accordingly. I therefore recommend the article for publication after minor revision.*
**Dear Reviewer, thank you for your detailed suggestions for accepting the manuscript with minor revisions. We have responded to the major comments and the minor comments throughout the manuscript. We are also thankful for the**
20    **comments on the ALE plots. Please find our responses below.**

**Major comments**

*Table 2 and lines 180-181: In all 12 cases the test $R^2$ is higher than the train $R^2$. In my understanding, test performance is usually below training performance. Higher test performance can occur occasionally, but it seems odd that this is the case for all your settings. Do you have an explanation for this? How did you split the train and test data set? Is there e.g. any possibility*
25    *of a bias due to the splitting? Did you try various ways of splitting the data? Related to this: You mention overfitting as an explanation of underestimation of year 2019 in line 328-329. But your better test performance compared to training perfor-mance would not indicate overfitting. (However, the following argument in line 330-334 seems reasonable.)* **Thank you for**

this comment. The training and testing sets are split completely randomly. For this purpose, the $slice\_sample$ command of the dplyr package is used with a seed (tested for different seeds as well as alternative random sampling commands) that randomly selects rows in the data (each row assembles the data for a specific time and space combination). The model is tuned using the training data (80% of the data). The test results shown in Table 2 were derived from predicting on the remaining 20% of the data, i.e., the test set. Since this set was randomly selected from the combined space-time space, it is reasonable to assume that the models are well suited for this realm but overfitted for either prediction in the isolated time or space dimension. As will be seen later when examining the year-specific out-of-sample predictions (lines 328-329), it appears to be the case that year-specific effects are not properly captured in the approach. This also shows potential difficulties of out-of-sample predictions of machine learning models such as random forest (l.327-328). However, we believe that such considerations should be the subject of further study. We added some information clarifying that the sampling is random based on time-space combinations (l. 149-150 and 181-182).

**Minor comments**

*[L. 65]: What are mean average effects? Mean and average seem synonymous.*

Thank you for this comment. ALE handles feature correlations by averaging and accumulating the difference in predictions across the conditional distribution (grid of realizations), thereby isolating the effects of the specific feature. The resulting ALE explanation is then centered around the feature's mean effect, so that the feature's main effect is compared to the average prediction of the data. For this reason, we used the phrase "mean average effects." However, we instead use average marginal effects here at this point.

*[Figure 1] Isn't it 2003 instead of 2004?*

Of course you are right, this was a typo. *[Table 2] If you explain the abbreviations T and P here, you should also explain SMI and SMIa for the sake of completeness.*

We added this information to Table 1. *L. 220-221] SMI4 (5th in non-cluster); SMI8 (6th in non-cluster)*

Thank you, we adapted this accordingly. *[Figure 4] Why is it not completely identical for "4a) no cluster" to the submitted version? SMI4 is now on fourth place instead of SMI12.*

As you can see in Figure A4 (appendix), the importance scores of SMI4 and SMI12 are very similar. To calculate the feature importance for a single feature, the model prediction loss (error) is measured before and after shuffling the feature values. The larger the increase in prediction error, the more important the feature was. The shuffling is repeated (here 50 times) to obtain more accurate results because the importance of permutation features is usually quite unstable. However, the permutation can still impact the final order. *[Figure 4] Mention in the caption that this is for PAM (4).*

Thank you, we added this information. *[Figure 4, A5-A9] You mention a grid size of 50, however in your answer to the reviews you state 100 (lines 139-148 of your rebuttal letter).*

We are sorry for the confusion. The ALE plots are based on a grid size of 50. *[L. 243] "A low drought signal of soil moisture is found for April (SMI4)." Odd phrasing, consider rephrasing.*

**Thank you, we use "A low drought signal based on soil moisture is noted for April (SMI4)" now.** *[L. 268] What about cluster 1 and 3? They also exhibit drought vulnerability. Maybe write "subregion such as cluster 2..."*

**We now use the recommended phrasing.** *[L. 270 and 338] crop yield potential*

65 **Thank you, we now use crop yield potential.** *[Figure 5] Missing data in gray should be indicated in the caption.*

**We added this information in the caption.** *[Fig. A4-A6] What is CR12?*

**This is a carryover from the work names. For the paper we now use the name Rain12.** *[L. 405] It is mean absolute error, not average. This was already mentioned in the first revision*

**We apologize for repeating this error.**

70 **ALE Plots**

*The ALE plots are confusing because the contain a variety of small mistakes and inaccuracies. Basically I think you want to show (as indicated in lines 414-418) in these plots the 3 best cluster algorithms for cluster 2, 4 and 6 for the two soil moisture index configurations "Soil moisture for uppermost 25cm" and "Soil moisture index for both uppermost 25cm and entire soil column" (one of the plots (Fig. 4) is in the main text), right?*

75 **The wobbliness of the curves is the result of the detailed grid used to estimate the effect of the predictor variables on the target variable. In addition, it may be an artifact due to the different effect of a feature in different subregions, which is not adequately accounted for by clustering. For this reason, we added the nonlinear smoothing functions. With a less detailed grid, the ALE plots would also be smoother, but could hide relevant information. The goal here is to show how the average marginal effects depend on the underlying clustering. For example, depending on the number of clusters**

80 **used and the resulting size of the clusters, information may or may not be revealed. This is noted, for example, in lines 292 - 293.** *[L. 376-378] This text was not updated. It is no longer PAM (8).*

**Thank you, we updated the text accordingly.** *Fig. A5 is not for PAM with 2 clusters as indicated in the main text and the caption, but identical to Fig. A6.*

**Thank you, something went wrong when setting the reference in latex. Also, as indicated later, Fig. A5 should represent**

85 **HIERARCHICAL(2) for the configuration of the soil moisture index, considering only the top 25 cm.** *Figs. A5-A9 all say "for both cluster". In three of these figures there are more than two. Also the plural-s is missing.*

**Thank you, we implemented your suggestions accordingly.** *First line of the caption: Instead of "SMI" call this "soil moisture index configuration" in all ALE plots as you do in Table 2.*

**Thank you, this is now implemented.** *Fig. A5 first line of the caption: Shouldn't this be HIERARCHICAL (2) instead of PAM*

90 *(2)? According to Table 2 HIERARCHICAL (2) is the best algorithm cluster size 2 and soil moisture index configuration "soil moisture for the uppermost 25 cm." Also in Line 414-415 it is mentioned as HIERARCHICAL (2).*

**Thank you very much for this comment. Indeed, the best model in terms of test R-squared is HIERARCHICAL(2)** *Fig. A7 second line of the caption: 2 clusters, not 6.*

**We changed this accordingly.** *Figs. A5-A9 first sentence of the caption: This is a bit misleading (in all ALE plots). It should*

95 *rather say something like: best combination of cluster algorithm (PAM) and the soil moisture index configuration "soil moisture*

*for the uppermost 25 cm" for cluster size 2. Each plot shows a different cluster size, so it is not part of the best combination, but rather it is the best combination for a given cluster size.*

**Thank you. We changed the first sentence of the caption as is stated in the following for Fig.A5: "Accumulated local effects (ALE) plots for the best combination of cluster algorithm and cluster size (HIERARCHICAL with 2 clusters) for a soil moisture index configuration that only considers the uppermost 25 cm." For the other captions this sentence is adapted accordingly.**

**Typos**

*[L. 115] extra bracket*
*[L. 233] bracket missing*
*[L. 240] 4d*
*[L. 249] Heat8 in upper case*
*[L. 247] bracket missing*
**Thank you, we have corrected the typos mentioned above.**

**Reviewer 2**

**General comments**

*The authors apply a random forest procedure to explain observed yield anomaly in Germany thanks to meteorological predictors. The central result of this paper is the quantification of individual non-linear contributions of meteorological variables/indices and especially the important role of soil moisture. The ALE plots are valuable material in this study (I am not sure about the interpretation of confidence intervals though, see specific comment n.3). To my opinion, this paper deserves publication after minor revisions.*

**Dear Reviewer, Thank you for your detailed summary and suggestions for accepting the manuscript with minor revisions. We have responded to the specific comments and technical corrections throughout the manuscript. We also thank you for the comments on the labels of the ALE plots. Regarding the ALE plots, there is currently no standard routine for uncertainty measures of the ALE plots (as described in more detail later). Please find our answers below.**

**Specific comments**

*[Fig 1b] You mention the identification of significant trends, did you perform some trend test or shift test?*
**Yes, we tested for trend in both the linear as well as in the LOESS model. Both models show significant trends.** *[L. 141] Is the clustering performed on raw yield directly, or anomalies? How should one interpret the clusters obtained: are the counties gathered in terms of yield magnitude or variability or occurrence of extremes?*
**Thank you for this comment. As stated in the text in lines 151 - 155, we use for clustering "monthly averages and daily**

observations of the meteorological data for the entire year. Soil moisture index is included for both the upper layer and the entire soil column. Average yields are also taken into account in the data for cluster formation. This is based on the intuition of taking into account time-invariant factors of each cluster that affect yields such as soil quality and average farm size. These factors are not considered in the random forest due to use of yield anomalies." Thus, the clusters aim to capture time-invariant factors, while the yield anomalies and corresponding ALE plots aim to capture the effects of extremes. *[Figure 4] the confidence intervals you obtain are related to the smoothing function, and not the robustness of the RF model itself. I am wondering to what extent it is possible to interpret it as an uncertainty of the variable effect (ex. l.238). To my understanding, this confidence interval tells us about the uncertainty of the smoothed curve, but not about the uncertainty of the local effect. (small remark: it is nice to specify the package for the ALE).*

Yes, the confidence intervals shown are based on the estimated LOESS functions and mostly refer to the number of realizations available to fit the function. Regarding the representation of uncertainty in the ALE representation, we would like to refer to an open issue in "General Pitfalls of Model-Agnostic Interpretation", an article by Molnar et al. (2020), the developer of the R package iml used for the analysis here: "While Moosbauer et al. [73] derived confidence bands for PDPs for probabilistic ML models that cover the model's uncertainty, a general model-agnostic uncertainty measure for feature effect methods such as ALE [2] and PDP [30] has (to the best of our knowledge) not been introduced yet." The package used and the corresponding command are presented in the caption of Fig. 4.

*[L. 405] Do I understand correctly that "the feature is shuffled" means that the variable, for which we want to compute the importance, is shuffled?. I.e., to get the importance of e.g. SMI3, the RF is re-run on the exact same data, except for SMI3, which is shuffled in time, and then the MAE is computed?*

Yes, you are right. To calculate the feature importance for a single feature, the model prediction loss (error) is measured before and after shuffling the feature values. Shuffling the feature values destroys the association between the outcome and the feature. The larger the increase in prediction error, the more important the feature was. The shuffling is repeated (here 50 times) to obtain more accurate results because the importance of permutation features is usually quite unstable.

**Technical corrections**

*[L. 233] missing ")".*
*[L. 240] error in the reference to the figure*
*[L. 257] missing ")"*
Thank you, we have corrected the technical corrections mentioned above.

It seems like there are some confusions in the figure captions:

*[Caption figure 3] not "observed and predicted data for the two clusters" but "four clusters"?*
Thank you, we corrected this accordingly. *[Figure 5] you chose to conduct the analysis with PAM 4 clusters, so in Fig5 it is four and not eight subregions, right?*

**Yes, you are right. We corrected this.** *[Figure A5 and A7] the captions don't seem to fit with the figures (clustering method and number of clusters).*

**We have changed the labels. For a more detailed explanation of what has been changed, we would like to refer you to the ALE section of Reviewer 1. We thank you very much.** *[L. 378] aren't the results in the appendix about PAM 4 and 6, and hierarchical 2 and 6?*

**We show the ALE plots for the best combinations of cluster algorithm, cluster size, and SMI for a defined soil depth (either the top 25 cm or both the top 25cm and the total soil column) that are not presented in the main text. Those are for the uppermost 25 cm soil moisture index configuration HIERARCHICAL(2) and HIERARCHICAL(6), and for the combination of top 25cm and total soil column HIERARCHICAL(2), PAM(4), and PAM(6).**

**References**

Molnar, C., König, G., Herbinger, J., Freiesleben, T., Dandl, S., Scholbeck, C. A., Casalicchio, G., Grosse-Wentrup, M., and Bischl, B.: General Pitfalls of Model-Agnostic Interpretation Methods for Machine Learning Models, arXiv.org, http://arxiv.org/abs/2007.04131, 2020.

170